# Controlling nanochannel orientation and dimensions in graphene-based nanofluidic membranes

Muchun Liu [1,2,3], Paula J. Weston[4] & Robert H. Hurt [1✉]

There is great interest in exploiting van der Waals gaps in layered materials as nanofluidic channels. Graphene oxide (GO) nanosheets are known to spontaneously assemble into stacked planar membranes with transport properties that are highly selective to molecular structure. Use of conventional GO membranes in liquid-phase applications is often limited by low flux values, due to intersheet nanochannel alignment perpendicular to the desired Z-directional transport, which leads to circuitous fluid pathways that are orders of magnitude longer than the membrane thickness. Here we demonstrate an approach that uses compressive instability in Zr-doped GO thin films to create wrinkle patterns that rotate nanosheets to high angles. Capturing this structure in polymer matrices and thin sectioning produce fully dense membranes with arrays of near-vertically aligned nanochannels. These robust nanofluidic devices offer pronounced reduction in fluid path-length, while retaining the high selectivity for water over non-polar molecules characteristic of GO interlayer nanochannels.

[1] School of Engineering, Brown University, Providence, RI, USA. [2] Department of Chemistry, Brown University, Providence, RI, USA. [3] Department of Civil and Environmental Engineering, Massachusetts Institute of Technology, Cambridge, MA, USA. [4] Department of Pathology and Laboratory Medicine, Brown University, Providence, RI, USA. ✉email: robert_hurt@brown.edu

Two-dimensional (2D) materials assemble into stacks with defined interlayer spaces, which can be widened by molecular intercalation or pillaring to produce nanoscale or sub-nanoscale channels. These well-defined slit-shaped pores can be exploited in emerging nanofluidic technologies, that include molecular and ionic separations, hydraulic-electric energy conversion and supercapacitors[1–5]. Among the many materials with intrinsic lamellar structures, graphene oxide (GO) has received the most attention, primarily as a selective membrane with sub-nanometer channels formed by the pillaring action of oxygen-functional groups on the nanosheet faces[6–8]. GO van der Waals (vdW) films with interlayer spacings approximately 8 angstroms are capable of precise molecular sieving[9] and can be readily fabricated over large areas by a variety of water-based coating or printing processes[1,10].

Conventional GO filtration membranes consist of stacked micron-scale nanosheets that align horizontally during evaporation/filtration assembly due to self-exclusion and substrate templating, which in turn aligns the associated nanochannels perpendicular to the desired flow direction through the membrane. The resulting fluid transport pathways are highly torturous with total lengths scaling as $L \approx t_{membrane} \times \left(\frac{L}{d}\right)_{nanosheet}$ where $t_{membrane}$ is the nominal membrane thickness and $\left(\frac{L}{d}\right)_{nanosheet}$ is the aspect ratio of the constituent nanosheets[11–13]. Typically monolayers have very high $\left(\frac{L}{d}\right)_{nanosheet}$ values, making permeate path lengths orders of magnitude greater than the membrane thickness, and providing a serious throughput limitation to liquid phase applications such as ultrafiltration and desalination[11,12,14]. Another important limitation of conventional GO membranes is excess water absorption and swelling[7], which can increase interlayer spacing from approximately 0.8 nm to as much as 6 nm, resulting in loss of molecular selectivity and mechanical stability[15].

One approach to overcome the flux limitation is to realign the nanosheets to create vertical (Z-directional) transmembrane nanochannels with path-lengths comparable to $t_{membrane}$. Several methods have been demonstrated to create vertical graphene structures for applications such as field emitters[16], capacitors[17], or edge-rich antibacterial surfaces[18]. Copper-assisted chemical vapor deposition has been used to directly grow vertical graphene[19], and GO nanosheets have been vertically aligned through ice-growth-directed assembly[20] and magnetic field alignment[18]. These methods successfully create vertical graphene architectures, but are not designed to create fully-dense, pore-free membranes in which transport occurs exclusively through 2D interlayer nanochannels, which is a requirement to achieve molecular selectivity. Another method to obtain vertical GO nanochannels is to tilt an entire planar GO film by 90 degrees. Such simple tilting, for a 1 μm thick GO film, produces only a single 1 μm wide slit rather than a membrane, and hand stacking of multiple micrometer-slits has been used to create novel devices for study of nanopore transport and molecular sieving[14]. Those devices are not thin membranes, but rather nanochannel-imbedded blocks with diffusion path lengths in the mm size range that are primarily useful for scientific experiments.

This work introduces a self-assembly method to create Vertically Aligned Graphene Membranes (VAGMEs). The fabrication concept originates from recent studies of textural control in graphene films, which exploit surface instability of planar graphene thin films on soft substrates under compression to produce wrinkled, crumpled, and mixed-mode surface deformation patterns[21–24]. Imbedding the zig-zag graphene structures in epoxy, with careful control of bubble formation, creates fully dense structures that can be microtomed into thin membranes to produce VAGME devices whose structure and membrane

performance are described below. These robust nanofluidic devices provide pronounced reduction in fluid path-length, while retaining the high selectivity for water over non-polar molecules characteristic of GO interlayer nanochannels.

## Results

**Fabrication and morphology of near-vertically-aligned GO membranes.** GO films are cast onto pre-stretched polystyrene substrates below their glass transition temperature, then heated to release the pre-strain and induce 2D isotropic crumpling in the stiff graphene coating. By physical fixation of the substrate on two of the four sides, the compression can be limited to one direction, creating hundreds of unidirectional periodic wrinkle patterns[21,22]. The individual wrinkles do not have smooth sinusoidal geometries, but rather nearly flat side walls and sharp (high-curvature) ridge tips[21,25]. An ideal triangular zig-zag structure (see Fig. 1a) has a total film length in cross-section that is longer than the base substrate length by a factor of $\frac{1}{\cos\theta}$, where $\theta$ is the rise angle from the substrate plane (Fig. 1a). This angle is directly related to the substrate relaxation ratio $\frac{L}{L_{initial}}$, $\theta = \cos^{-1}\left(\frac{L}{L_{initial}}\right)$ and the transport path lengths $t$ across the membrane become $\frac{t_{membrane}}{\sin\theta}$ (Fig. 1a). At high extents of compression (low $\frac{L}{L_{initial}}$ values) the side walls are forced to tilt at high rise angles to the substrate plane and thus adopt a strong vertical component that forms the basis for the near-vertical nanochannel arrays. Figure 1b shows the detailed VAGME fabrication process, which begins with conventional drying-induced assembly of planar GO nanosheet vdW films on substrates to produce dense, uniform arrays of nanochannels (Fig. 1b i, GO nanosheets are characterized as in Supplementary Fig. 1). The nanochannels are then partially reoriented by choosing pre-stretched polystyrene for the substrate, which undergoes thermally activated shrinkage to compress the GO film and create 1D wrinkles (Fig.1b ii). We observed that neat (metal-free) wrinkled GO films contain voids at the interface of the film and bottom substrate, and the wrinkle deformation is irregular, producing chaotic microstructures seen in cross-section (Supplementary Fig. 2a). Increasing the mechanical strength of GO films (by increasing thickness from 1 to 2 μm) did not completely eliminate the distorted deformation or the voids. We hypothesized that increasing the substrate-film interfacial adhesion would prevent local delamination and eliminate the voids thus improving wrinkle uniformity. Since both the GO nanosheets and the air-plasma-treated polymer surface are negatively charged, we explored metal cation addition as an electrostatic bridge between film and substrate. After considering a range of candidate cations through experiments and modeling (see Supplementary Table 1, Supplementary Fig. 2 and associated discussion), $ZrOCl_2$ was chosen as the optimal additive to the deposition suspension. The ZnO(II) cation binds successfully to GO and reverses the zeta potential from $-48$ to $+40$ mV, and resulted in more adherent films and more regular wrinkle textures (Supplementary Fig. 2b–d). The colloidal behaviors of metal ion-GO suspensions were explored using a model of cation complexation and electrostatic charge screening, and the model predicts that the effective surface charge flipping of in Zr-GO suspensions was achieved by the very high stability constant of the ZrO(II)-carboxylate complex (see detailed discussion in Supplementary Note 1).

The wrinkled Zr-GO films are then removed from the relaxed substrate and imbedded in an epoxy matrix (Fig. 1b iii). Due to high viscosity and hydrophobicity, epoxy resin cannot diffuse into GO gallery spaces[14] (which would cause undesirable blockage of nanochannels), but successfully impregnates the microstructural

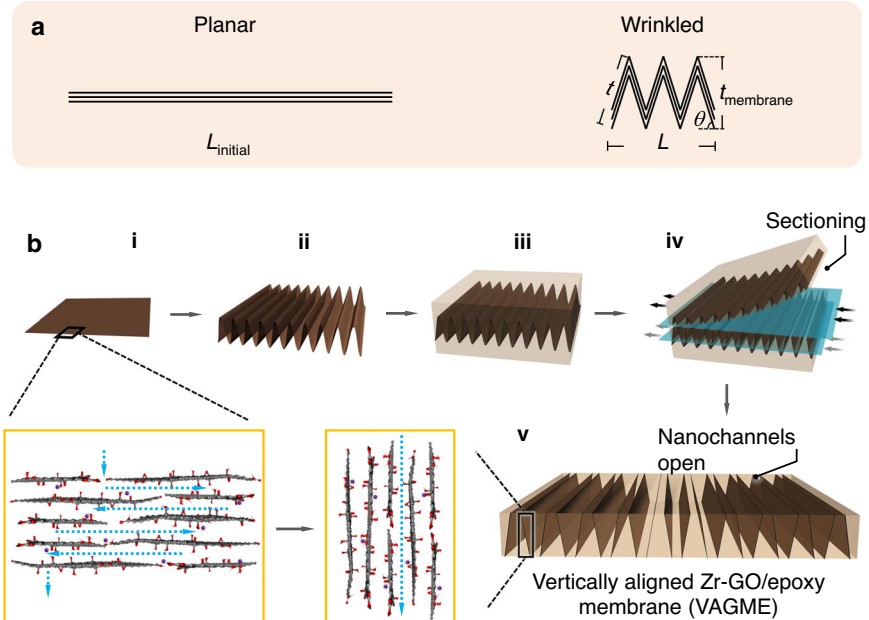

**Fig. 1 Structure and fabrication steps leading to vertically aligned Zr-GO/epoxy membranes. a** Sketch of compressive film wrinkling. Side view of planar and 1D wrinkled films. Figure illustrates that wrinkling tilts the horizontal line into multiple near vertical line segments. **b** Fabrication steps leading to vertically aligned Zr-GO/epoxy membranes. **i**, Drying-induced assembly of Zr-GO nanosheets (Zr/C atomic ratio approximately 1/22) on pre-stretched polystyrene substrate (GO film thickness 1 μm). Inset: nanostructure of planar Zr-GO films with horizontal alignment and tortuous flow pathways. Brown strips represent 1 μm thick multilayer GO film segments. The light-yellow box represents the epoxy matrix. Purple spheres represent ZrO(II) cations (unhydrated state for reference). Fully hydrated ZrO(II) diameter is approximately 1 nm, implying that ZrO(II) likely exists in the interlayer spaces in a partially hydrated state complexed with O-containing groups on GO. **ii**, Wrinkled Zr-GO films are produced by thermally activated mechanical compression. **iii**, Wrinkled Zr-GO films are removed from the substrate and imbedded into epoxy resin. **iv**, Multiple cycles of microtome sectioning yield Zr-GO/epoxy composite membranes. **v**, Side view of vertically aligned Zr-GO/epoxy membrane (VAGME) with the entrances to interlayer nanochannels open at the top and bottom surface. This method transforms a single planar Zr-GO film into hundreds of vertical film segments, where each segment is an array of approximately 1000 GO interlayer nanochannels aligned with a strong Z-directional component.

gaps around the wrinkle features and mechanically stabilizes the structure. The wrinkled Zr-GO/epoxy composite is then sectioned by microtome using heavy duty high profile stainless blades to remove the ridge tips on the top and bottom of the zig-zag film, exposing nanochannel entrances and exits (Fig. 1b iv). A challenge was to develop a microtome sectioning technique that does not introducing holes or cracks that would act as short circuits for molecular transport through the membrane (see Supplementary Fig. 3 and associated discussion). The final product – a vertically aligned Zr-GO membrane (VAGME) – is shown in Fig.1b v, where the original planar film has been realigned and sectioned into independent transmembrane film segments, each segment consisting of an array of GO interlayer nanochannels at 100 arrays per mm.

Figure 2 presents detailed morphological characterization of the VAGME microstructures at different stages of fabrication. Pre-stretched polystyrene rectangles of $11.0 \times 3.0$ mm$^2$ thermally relax to $3.0 \times 3.0$ mm$^2$. The unidirectional relaxation ratio of $\cos\theta = \frac{3}{11}$ corresponds to $\theta = 74$ degree rise angle in the ideal zig-zag geometric model (Fig. 1a). Observed rise angles are in the range 60–80 degrees (Fig. 2) and the thickness of the films increases from 1 μm to about 100 μm. Figure 2a shows dark brown-colored, wrinkled Zr-GO films imbedded in an epoxy cylinder, with is observed to be free of pores or cracks by microscope observation. After thin sectioning, the VAGME is a free-standing and robust film that reveals a pattern of parallel dark strips in the transparent (epoxy) matrix (Fig. 2b). The top view of the wrinkled Zr-GO films shows a corrugated texture with continuous 1D wrinkles of 3.0 mm in length, and approximately 20 μm in wavelength (Fig. 2c). Under SEM the VAGME shows a

smooth surface with a pattern of approximately 300 fine strips, each strip representing the terminus of a near-vertical Zr-GO film segment of approximately 1 μm in width. The high magnification view (Fig. 2d and Supplementary Figs. 4-5) of an individual Zr-GO strip shows a tightly packed, multilayered structure, which appears identical to original planar Zr-GO films. More information on the assembly and alignment of nanosheets in both the original GO and Zr-GO films can be found in the Supplementary Note 2, 3 and references therein. The side view of wrinkled Zr-GO films in the matrix shows a zig-zag pattern with wrinkle amplitude approximately 50 μm (Fig. 2e). The final VAGME is 20 μm in thickness after removing the ridge tops to access the GO nanochannels and convert the original continuous film to a set of near-vertical Zr-GO segments (Fig. 2f).

**Selective molecular transport through VAGME nanochannels**. A major challenge in designing and fabricating near-vertically aligned nanochannel membranes is to ensure that molecular transport occurs only through the nanochannels and cannot bypass the nanochannels through membrane cracks or pores. Parallel transport through such microscale membrane defects would be rapid and non-selective, thus negating the primary benefit of the nanochannel technology. Potential defects in VAGMEs include (i) trapped air bubbles in the epoxy, which become holes in thin sections, (ii) cracks in the GO films, and (iii) delamination openings at the epoxy-GO interfaces occurring during curing or processing.

Microscopic inspection of our final VAGME membranes did not reveal visible micro-scale defects. This inspection technique is not reliable for identifying defects <100 nm, however, which can

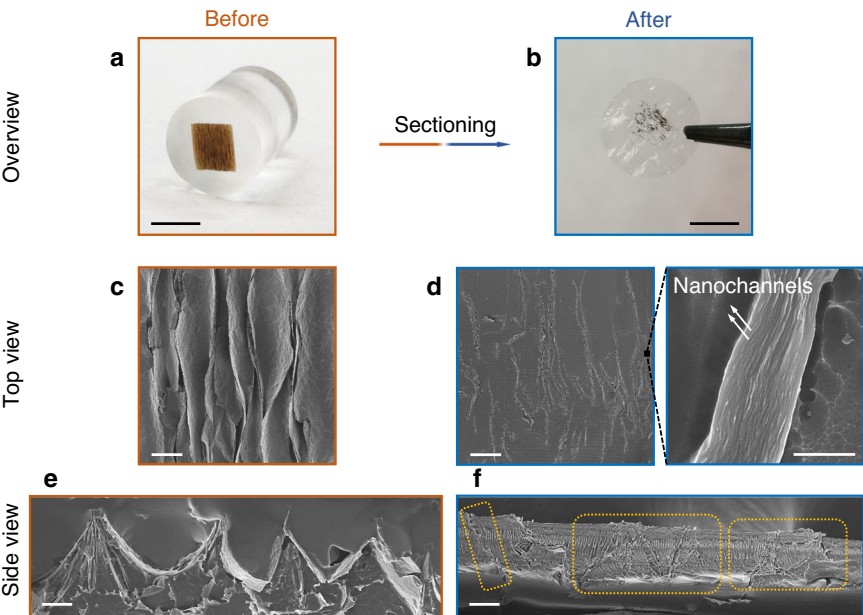

**Fig. 2 Morphologies of wrinkled Zr-GO films and VAGME during fabrication. a**, **b** Photographs of wrinkled Zr-GO films imbedded in epoxy matrix and, after thin sectioning, a free-standing VAGME. Scale bar, 4 mm. **c**, **d** Top views of wrinkled Zr-GO films and a VAGME. Wrinkled Zr-GO films show corrugated textures. VAGME shows a strip pattern associated with GO film segments whose edges intersect the top surface. Scale bar, 100 μm. High-magnification view of one Zr-GO strip on the VAGME surface, showing a uniform, close-packed array of nanosheet layers and interlayer nanochannels. Scale bar, 1 μm. **e** Side view of wrinkled Zr-GO/epoxy composite, exhibiting regular zig-zag patterns as desired. **f** Side view of VAGME, showing several individual Zr-GO near-vertical segments after ridge removal by microtome (highlighted). Scale bar, 20 μm.

also degrade membrane selectivity. We therefore used selective molecular vapor permeation as a probe for the presence or absence of nanoscale defects. A distinctive behavior of GO interlayer nanochannels is their ability to pass water vapor rapidly, while excluding all non-polar and many other polar molecules[3,9,26], including hexane[27]. In the absence of membrane defects, transport will occur only through nanochannels with very high permeation rates for water vapor and near perfect exclusion of hexane vapor. In contrast, defects significantly larger than the sub-nm interlayer channels would pass hexane and water at similar rates (for similar vapor pressures) governed by their similar gas-phase diffusion coefficients. Therefore, single-phase permeation experiments with water and hexane vapor were carried out to test the hypothesis that transport in these membranes occurs primarily through GO interlayer nanochannels rather than defects. To provide additional information, we included vapor permeation of 2-propanol, water/2-propanol mixtures, and performed experiments on both VAGME films and epoxy-only controls.

Figure 3a shows the test apparatus[27] which is heated to produce a range of vapor pressures up to 1 bar corresponding to the normal boiling points of water (100 °C), hexane (68 °C) and 2-propanol (82 °C). Magnetic stirring is introduced to reduce mass transfer resistance on the upstream side, and the device was operated in a fume hood to provide convective flow over the VAGME surface to reduce downstream mass transfer resistance. Water vapor transmission rates through VAGME films show a linear dependence on upstream water vapor pressure and approach 10 mmol mm$^{-1}$ hr$^{-1}$ at 1 bar vapor pressure, while (pure) hexane or 2-propanol permeation is at or below the detection limit (Fig. 3b). Pure epoxy films (20 μm thick), used here as a negative control sample, show no measurable flux for all three solvent vapors indicating flaw-free full density (Fig. 3c).

The vapor transport behavior associated with water/2-propanol liquid mixtures is shown in Fig. 3b. In this liquid mixture vapor, VAGME allows water permeation but excludes 2-propanol

molecules. As shown in Supplementary Table 2 and Fig. 3b, the flux of water or 2-propanol in a mixture is comparable with that of individual vapor. In general, water molecules enter nanochannels driven by capillary forces, and are reported to flow preferentially through non-oxidized regions[3,12,28]. The size of these channels can be estimated by subtracting the width of a pristine (non-oxidized) graphene layer (0.34 nm) from the XRD interlayer spacing (here 0.88 nm) to give 0.54 nm. Some transport processes may occur through the oxidized regions, where channel sizes are smaller due to out-of-plane oxygen-containing functional groups[15,29,30] and can be estimated here as 0.34 nm (see detailed discussion in Supplementary Note 4). Both types of channels are expected to reject 2-propanol molecules, which have a (bare) kinetic diameter of 0.49 nm[31] and are expected to be larger within the water-saturated channels in these membranes due to hydration at the site of the H-bond active hydroxyl group. The rejection of hexane occurs primarily through its hydrophobicity (water insolubility)[9]. The data clearly show that VAGME films exhibit the characteristic high selectivity of GO nanochannels for water over hexane or 2-propanol, and confirm the absence of pores and interfacial cracks that would degrade this selectivity. Microscopic inspection of the VAGME films after the vapor permeation experiments showed no structural changes, indicating thermal stability up to 100 °C, unlike conventional GO films which have a tendency to crack at temperatures above 40–60 °C[9,27]. Further, the width of the nanochannel arrays that intersect the top VAGME surface were observed to be unchanged by the permeation experiments, which suggests these membranes are swelling-resistant due to the confining effect of the rigid epoxy matrix[14]. Overall, VAGMEs offer pure GO nanochannel transport in a platform that is mechanically stable, thermally stable up to 100 °C, and swelling resistant.

Table 1 gives comparisons of water vapor flux through VAGME vs. conventional GO membranes[9,27]. At 60 °C, VAGME water vapor flux is similar to that of conventional GO membranes on a total area basis, even though only a fraction of the total

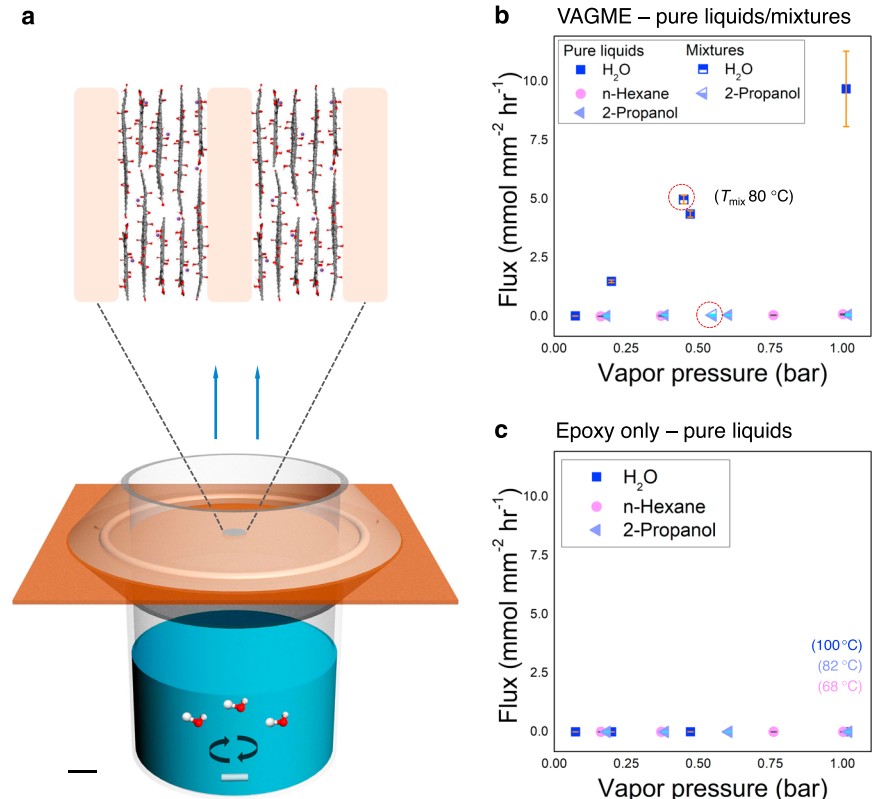

**Fig. 3 Measurements of selective molecular transport through VAGME nanochannels. a** Custom diffusion cell for temperature-dependent vapor permeation experiments. VAGME films are adhered on the top of a copper sheet spanning a 4 mm diameter hole by epoxy adhesive, and the plate is clamped and sealed on top of a stirred liquid/vapor cell. Scale bar, 5 mm. **b** Measured vapor fluxes through VAGME films for pure liquids and liquid mixtures. Pure liquids: water, n-hexane and 2-propanol. Liquid mixtures (circled in red dot line): water and 2-propanol. **c** Control experiments for water, n-hexane or 2-propanol vapor permeation through epoxy-only membranes of same (20 μm) thickness. Flux values are normalized by the active Zr-GO (nanochannel array) area.

| Table 1 Water vapor fluxes measured through VAGME devices and conventional GO films. | | | |
|---|---|---|---|
| Membrane | Flux (60 °C) kg m$^{-2}$ hr$^{-1}$ | Flux (80 °C) kg m$^{-2}$ hr$^{-1}$ | Flux (100 °C)** kg m$^{-2}$ hr$^{-1}$ |
| Conventional GO films | 1.7–4.1* [9,27] | (film cracking) | (film cracking) |
| VAGME (total area basis) | 1.9 ± 0.1 | 5.6 ± 0.2 | 13 ± 2 |
| VAGME (active area basis) | 27 ± 1 | 79 ± 2 | (1.8 ± 0.3) × 10$^2$ |
| | **Normalized flux (60 °C) kg m$^{-2}$ hr$^{-1}$ μm** | **Normalized flux (80 °C) kg m$^{-2}$ hr$^{-1}$ μm** | **Normalized flux (100 °C) kg m$^{-2}$ hr$^{-1}$ μm** |
| VAGME (active area basis) | (5.4 ± 0.2) × 10$^2$ | (1.6 ± 0.1) × 10$^3$ | (3.5 ± 0.6) × 10$^3$ |

*The flux value from ref. [9] includes a calculated temperature correction based on vapor pressure driving force.
**The 100 °C data is time-dependent and the table entries are average fluxes over 12 h.

membrane area is occupied by nanochannel arrays. Renormalizing fluxes by active (GO) area, as determined by SEM analysis of top surface images, shows a VAGME enhancement factor of approximately 16. Normalizing flux values by membrane thickness, which differs significantly across the comparison cases, gives even higher enhancement factors. VAGME fluxes increase at higher temperature and vapor pressure, but comparisons with conventional membranes are not possible due to their tendency to crack. VAGME films in contrast show are intact and show high water vapor transmission rates during contact with 80 or 100 °C water vapor over 12 h. Because GO is susceptible to thermal reduction, the 100 °C experiments were repeated with time-resolved flux measurements (Supplementary Fig. 7a). These data show a slow decline in water vapor flux from approximately 4.5 × 10$^3$ to 1.6 × 10$^2$ kg m$^{-2}$ hr$^{-1}$ μm over 24 h. We believe this is due to slow hydrothermal reduction (deoxygenation) of GO which is known to reduce water permeability[32], due to interlayer contraction and loss of hydrophilicity. XPS results confirm a progressive deoxygenation process (Supplementary Figs. 7–8). Interestingly, there is no significant change in interlayer spacing during this process (Supplementary Fig. 7), as normally occurs in the GO to reduced-GO transition, despite the very significant degree of deoxygenation. We suspect this is due to the intercalated cations, which continue to pillar the GO nanosheets as oxygen is removed[12]. We believe the primary mechanism for the drop in water vapor transmission rate is reduction in hydrophilicity with progressive deoxygenation. Beyond vapor permeation, liquid filtration tests on NaCl and MgCl₂ aqueous solutions were also conducted and discussed. More detailed discussion is included in the Supplementary Note 4.

Overall, the key concept in this study is to geometrically deform a planar film, in a single processing step, to produce

hundreds of tilted short film segments, where each segment represents an array of approximately 1000 near-vertical nanochannels for rapid, direct (non-tortuous), through-membrane transport. The technique accomplishes a 300-fold increase in active area (face area occupied by the entrances to nanochannel arrays) relative to the simple hand tilting of a complete planar GO film. The current proof-of-principle demonstration uses a tilt angle of 74 degrees to produce an active area fraction of 10%, but this may be improved through further work at higher compression ratios (higher tilt angles) possibly using cycled contraction[33]. There is no known fundamental limitation on improvement by this method, and the active area fraction theoretically approaches unity as tilt angle approaches 90 degrees (the ideal tightly folded membrane whose surfaces consist entirely of graphene nanosheet edges). Improvements may also be expected through alternative wrinkling approaches based on stamping or grating templates, or new de-capping methods that include chemical etching as alternatives to mechanical sectioning[34].

In summary, the emerging field of 2D nanofluidics requires new methods to create well-defined arrays of interlayer nanochannels with precise control of width, length, and orientation. Here we applied recent techniques for compressive texturing of 2D thin films to a unique Zr-GO composite to direct the self-assembly of near vertically aligned nanochannel arrays with uniform and controlled fluidic pathlengths. The self-assembled structure can be effectively captured by epoxy imbedding and converted into mechanically robust microscale membranes, where molecular transport is allowed only through Z-directional, transmembrane GO interlayer channels of defined length. This approach simultaneously addresses several well-known limitations in current GO nanofluidics related to undesirable (horizontal) channel orientation, thermal instability, and uncontrolled water swelling that degrades molecular selectivity. We anticipate future work will focus on exploiting this approach to create devices for specific technological applications, including stable selective molecular sieve membranes for liquid phase separations such as ultrafiltration and reverse osmosis.

## Methods

**Materials**. Ethanol, zirconyl chloride octahydrate ($ZrOCl_2 \cdot 8H_2O$) and methylene chloride were purchased from Sigma-Aldrich. Hexane was purchased from Fischer Scientific. Thermally responsive polyethylene heat shrink films were purchased from Grafix, and Epofix from Electron Microscopy Sciences. Copper sheets (0.3 mm thick) were purchased from McMaster Carr. All water was deionized (18.2 M$\Omega$, milli-Q pore). All reagents were used as received without further purification.

**Preparation of wrinkled Zr-GO films on polystyrene substrate**. GO nanosheets were prepared by a modified Hummers' method. Detailed synthesis and characterization of GO are in Supplementary Note 1. The Zr-GO suspensions used in the colloidal and film formation experiments contained 3.55 mg mL$^{-1}$ GO and 8 mM $ZrOCl_2$ aqueous solution. The polymer shrink film was cut into $60.0 \times 11.0$ mm$^2$ rectangles and washed with ethanol. Once dry, samples were treated with air plasma in a Deiner Atto standard plasma system with a borosilicate glass chamber and a 13.56 MHz, 0–50 W generator. The chamber pressure was pumped down to and maintained at 0.13 mbar while being flushed with air for 5 min. Plasma was then generated at 100% power (50 W) for 15 min followed by slow venting of the chamber. Each polystyrene rectangle is masked with scotch tape only to leave a $3.0 \times 11.0$ mm$^2$ clean gap in the center, which is to limit subsequent drop casting in selected area. Then 16.8 μL of Zr-GO suspension is drop cast on the gap and form a 1 μm Zr-GO coating at 60 °C. The protection tape was then removed and both uncoated sides are clapped for 1D shrinking at 130 °C for 9 min. Finally, $3.0 \times 3.0$ mm$^2$ uniform wrinkled Zr-GO films are obtained in the center of the polystyrene substrate without distortion.

**Fabrication of wrinkled Zr-GO/epoxy composites and membrane sectioning**. 5.00 g Epofix embedding resin 1232 R and 0.56 g Epofix hardener 1232 H were mixed in an aluminum cup and under stirring for at least 3 min. Then the mixture was put into a vacuum chamber to exhaust bubble for 20 min. Approximately 20 μL mixture was dropped on the surface of wrinkled Zr-GO films and followed by another 20 min of degassing, which facilitates the impregnation of epoxy into micro gaps. The mixture was left in a desiccator for hardening overnight. Once cured, wrinkled Zr-GO/epoxy composite was put upside down, where the epoxy

side was adhered to the surface of an epoxy pillar. Then the polystyrene substrate was dissolved in methylene chloride for 15 min, please be aware that only the polystyrene part was immersed. After removal of polystyrene substrate, another step of Epofix impregnation was carried out on the exposed wrinkled Zr-GO films as mentioned before (Fig. 2a). This procedure is to ensure Zr-GO films are always fixed on a hard substrate during fabrication thus the structural integrity is maintained. Then, wrinkled Zr-GO/epoxy composite was thin sectioned using an automated microtome, the cut thickness is set to 20 μm. The active GO and total membrane areas are 0.9 and 13 mm$^2$ respectively.

**Pure liquid vapor permeation experiments**. Water, n-hexane or 2-propanol with a volume of 17 mL was pipetted into an open top glass vessel with O-ring flange. Test membranes were adhered over a 4.0 mm diameter hole in a copper sheet by Epofix, then hardened in a desiccator overnight. After curing, the copper plate was clamped in a two-piece vessel and sealed with vacuum grease coated rubber rings. Then the setup is put on a hot plate and heated to desired temperature under stirring at 200 rpm. The temperature inside the vessel is calibrated according to different liquids before setting up the hot plate. Water, n-hexane or 2-proponal permeation was then measured as gravimetric loss after 4–6 h and converted to average flux values.

**Liquid mixture vapor permeation experiments**. Water and 2-propanol with a volume of 5 mL respectively was pipetted into an open top glass vessel with O-ring flange. Test membranes incorporated copper plate was clamped in a two-piece vessel and sealed with vacuum grease coated rubber rings. Then the setup is put on a hot plate and heated to 80 °C under stirring at 200 rpm. Permeation of water/2-proponal mixture was then measured as gravimetric loss after 4–6 h and converted to average flux values. The individual flux of water or 2-propanol was calculated by combining refractive indices $n_D$ (linear relation with ratio of two molecules) and gravimetric loss of liquid mixtures, details are shown in Supplementary Note 4.

**Analysis of Zr-GO films exposed to 100 °C water vapor**. Planar Zr-GO films were prepared by drop casting 2 mL of Zr-GO suspension on a hydrophobic Teflon substrate, followed by peeling to obtain a freestanding membrane. The membrane is placed in a 130 °C oven for 9 min to ensure consistent treatment with the membranes subjected to the thermal compressive wrinkling process. The thermally treated Zr-GO membranes were cut into multiple smaller pieces and placed above 1 bar boiling water for various time. These treated Zr-GO films are then characterized by XRD and XPS.

**Liquid filtration tests**. A set of H-shaped glass cells (volume 50 mL) was employed for liquid filtration test. As shown in Supplementary Fig. 9, a VAGME was fixed between the feed and permeate compartments by two O-rings to provide a leak-free environment. 50 ml of 0.05 M salt (NaCl or MgCl$_2$) aqueous solution and nano-pure water were filled into feed and permeate compartments, respectively. All solutions were under stirring at 200 rpm and room temperature. The permeated salts were measured using a conductivity probe and calculated to concentration after calibration. Details are shown in Supplementary Note 4.

**Statistical analysis**. All vapor and liquid tests were run in triplicates, and the standard errors for the analytical results were used to generate and present error bars.

**Characterization**. The surface morphologies of graphene structures were investigated using a field emission scanning electron microscope (SEM) (LEO 1530 VP) operating at 10.0 kV for low-, medium- and high-resolution imaging. Before the SEM imaging, all samples were coated with a layer of AuPd (<1 nm). The phases of graphene films were identified by X-ray diffraction spectrometry (XRD) on a Bruker AXS D8 Advance instrument with Cu Ka radiation ($\lambda = 1.5418$ Å). Crystalline peaks for the Si substrate were removed. The chemical compositions of Zr-GO films were identified by K-Alpha X-ray photoelectron spectrometer (XPS). C1s peaks are fitted using XPSPEAK. Sectioning of Zr-GO/epoxy composites is carried on a Leica RM2265 automated microtome, equipped with C.L. Strukey, Inc. heavy duty high profile disposable microtome blades. The cut thickness is set to 20 μm. Retractive indices were read on a Fisher Scientific Abbe Model LR45302 Refract-ometer ($n_D$ resolution: 0.0001). Conductivity in liquid filtration tests was read on a Vernier Conductivity Probe LabQuest 2 (resolution: 0.1 μS cm$^{-1}$). Photographs were taken by an EOS digital SLR and compact system camera Canon EOS 100D.

## Data availability

The datasets generated during and/or analyzed during the current study are available in the Brown University Superfund Research Program Digital Archive at https://doi.org/ 10.26300/r4sv-a551. All other relevant data are available from the corresponding author upon request.

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

## Acknowledgements

The authors acknowledge financial support from National Institute of Environmental Health Sciences (NIEHS) Superfund Research Program P42 ES013660.

## Author contributions

M.L. and R.H.H. conceived this project; M.L. performed materials fabrication and permeation tests; M.L. and P.J.W. carried out microtome sectioning; M.L. and R.H.H. analyzed the data and wrote the paper. All authors provided scientific input, edited, and approved the final manuscript.

## Competing interests

The authors declare no competing interests.
