## [Peer Review File · Nature Communications]

REVIEWER COMMENTS

Reviewer #1 (Remarks to the Author):

The manuscript entitled "Controlling Nanochannel Orientation, Length, and Width in Graphene-Based Nanofluidic Membranes" describes the fabrication of a vertically aligned stacked graphene oxide membrane that is tested for gas separation.

This is a very well written paper that describes step-by-step the rationale of the process of making this vertically aligned GO membranes. The study was very comprehensive in addressing potential leaking issues, normalizing performance to active area and thickness, and describing change in performance over time. Therefore, this is a very complete work. It is also very novel. Horizontal stacked GO membranes have been used abundantly in the literature for different separation applications however this vertical design of the membrane has not been considered. The current manuscript covered well the literature on this topic, the limitations of horizontally stacked GO membranes, and how the vertical design address this issue.

I believe this paper can be of high interest to the field and inspire a new wave of GO-based membrane fabrication studies. However, to successfully do, I encourage the authors to consider a few elements of discussion that would significantly help how to take this work to the next steps.

Main comments:

Choice of Zr: The manuscript is a bit light on why Zr was chosen as the intercalation ion. It mentions, at lines 89-91 "It was found that Zr doping using the water soluble $ZrOCl_2$ as an additive to the deposition suspension produced more adherent films and more regular wrinkle textures (Supplementary Fig. 1b and 2)." Why was Zr chosen for intercalation? The SI goes into more details about the need for having positive charge to maintain colloidal stability, which makes sense. However, they also show data with Fe and show that Fe leads to higher gaps, suggesting that Zr is better. Why? Were the experiment with Fe done at the same zeta potential? Would the same results be obtained with Cu? Or Ca? This is an important aspect and could represent an important element to tune in the design. More discussion on the choice of the intercalation ions would be useful.

Thickness and active area: The study measure water vapor flux and normalized the flux to thickness and active area to show that the vertically aligned GO membranes had higher permeability than horizontally stacked GO membranes. For real life considerations, it is the permeance that matters though, and not the permeability. However, higher permeability can be very important IF there is a path to increasing the permeance, by making the membrane thinner or increasing the active surface. The SI discuss how the sample thickness was a major challenge in obtaining clean cuts without cracks and defects. Will this be a limitation that will prevent ever reaching permeance values that are of interest for real life applications? I suggest to include a paragraph of discussion of the potential avenues the authors see to improve the performance of the membrane and what are the challenges to do so.

Table 1 – error bars need to be added to the VAGME results.

Go synthesis: The paper says "GO nanosheets were prepared by a modified Hummers' method". However, modified Hummer's method varies a bit between papers in the ratios of reagents to graphite, temperature, reaction time, etc. The authors cite a paper that is from 2017. However, that paper says: "Graphite oxide (GO) was prepared by applying a modified Hummers method [21, 44]" without more information on the actual quantities and ratios of the material, taking us down the rabbit hole of hunting for the actual experimental details (now with two references to look for to try to figure out what was done). This is not acceptable – the manuscript should have all the experimental details needed to be able to reproduce the experiments as closely as possible. This can go into the SI.

Reviewer #2 (Remarks to the Author):

In this manuscript, Liu et al. reported the use of a self-assembly method to fabricate vertically aligned graphene membranes. Briefly, GO films are first cast onto a pre-stretched polystyrene substrate, which was then compressed in a horizontal direction to create unidirectional periodic wrinkle patterns in vertical direction. The wrinkled GO films were removed from the substrate and imbedded into epoxy resin, which could be sectioned into thin, composite GO/epoxy membranes. The fabricated membranes were characterized on SEM to verify the aligned structure of GO. Additionally, the authors demonstrate the molecular selectivity of membrane on a vapor permeation setup using water or hexane as feed water. Generally, the method using epoxy embedding, followed by microtoming to create vertical GO film is not novel, because a similar method has been reported in a previous study (Nair et. al, Nature Nanotechnology, 2017, DOI: 10.1038/nnano.2017.21). Such fact significantly lowers the novelty of this study. Additionally, the lack of systematic investigation and practical implications decrease the overall quality of this work, and some key conclusions in the manuscript are also not sound. Detailed comments are listed below:

1. In Figure 1, the authors illustrate the overall procedures and ideas on fabricating vertical GO film. As I mentioned above, it is not clear how their procedures are different from the previous study using the similar epoxy technique. They used a pre-stretched polystyrene substrate to create wrinkled GO films (Figure 1b). Such procedure is also not necessary for obtaining the vertical GO structure. For example, the epoxy embedding (figure 1c) could be directly applied to the deposited GO film. After that, the vertical alignment could be easily achieved by manually rotating the angle of the film in microtoming (Figure 1d).

2. According to Figure 1e, the authors considered their GO nanosheets with a uniform size (or at least small size distribution). However, most of the GO suspension has a much larger size distribution. As each channel is formed by two laboring nanosheets, the selective channel length is likely determined by the shorter GO sheets. Microtoming of the composite film might only expose the longer GO sheets. In this case, the majority of selective channels were still embedded in the resin, which could not exhibit any permeability.

3. The authors only provided some SEM micrographs (Fig. 2) to sustain their claim on the formed vertical GO structure. However, these rough images are good evidence to verify the nanosheet alignment. More importantly, through SEM observation, I can see that most of GO nanosheets were aligned with some titled angles, which is far from the authors' definition of "vertical alignment". Therefore, more systematic material characterization is needed to verify the alignment of GO.

4. The author claimed that "microscopic inspection of our final VAGME membranes did not reveal visible flaws" (line 163). It sounds like the authors considered the SEM morphology as the first evidence to demonstrate the integrity of their films. However, it should be noted that their SEM micrographs only allow the observation of structure on the micrometer scale (evidenced by the scale bars in the figures), which is at least three orders of magnitude larger than the selective GO-channel size (less than 1 nm). In other words, even some defects (<100 nm) were present in the film, it is challenging to observe on their SEM micrographs. But these defects were detrimental to the selectivity of the composite film. Therefore, in-depth material characterization is also needed to reveal the actual assembly behaviors of GO nanosheets.

5. Regarding the vapor permeation experiments, the authors used this method to demonstrate the integrity and selectivity of the GO membrane. There are a couple of defects in their experimental design.

a. First, membrane separation of water and hexane is not needed as they would form two phases in nature. The authors need to provide some background explanation on selecting these two solvents as feed water.

b. Second, even if the authors aimed to utilize the size difference in these two solvents to demonstrate the selectivity of their fabricated GO films, distinct differences in the physicochemical properties of two solvents would limit mechanistic elucidation. For example, physicochemical properties such as hydrophilicity/hydrophobicity and surface tension may contribute to the transport of these vapor in the composite film, which in turn affect the selectivity. The authors need to select more solvents as vapor feed for a comprehensive evaluation of membrane selectivity, thereby excluding the possible contribution of other properties. Additionally, the authors performed the vapor permeation experiments using feed with a single solvent (either water or hexane). At least mixed vapor of water and hexane should be used to verify their

claim on the selectivity.

c. Fourth, it is not clear how these vapors would transport through GO channels. Will they interact with GO nanosheets while diffusing through GO nanochannels?

d. Lastly, I am curious is there is any wetting of the GO film during the vapor permeance experiments. As GO is hydrophilic, it is likely that the water vapor condenses on the inner side of the film and gradually transport to the outer layer. In contrast, hexane vapor was repelled by the hydrated GO film, which may responsible for the observed non-flux of hexane in Figure 3b.

6. Instead of performing vapor permeation experiments, liquid filtration tests using different solute-solvent pairs should be a more facile way to demonstrate the stability and selectivity of their fabricated films, particularly considering their envision on the potential application of the films as "membranes for ultrafiltration and reverse osmosis" (Line 238).

7. According to the data in Table 1, the benefit of vertical GO in gaining water flux enhancement is negligible. For example, the vertical GO film exhibited comparable water flux to that of the conventional GO film (1.9 vs 1.7~4.1 kg m⁻² h⁻¹). This result implies that the decrease of tortuosity, which resulted from the vertical alignment, is not that significant compared to the non-aligned GO film. Even the authors argued to use effective areas for calculation, the water flux of the vertical film was enhanced to 27.1 kg m⁻² h⁻¹. Such water flux is still only around 6 times larger than that of the conventional film. More importantly, using effective area calculation does not sound as the epoxy is always needed for fabricating these vertical film (I would assume making epoxy-free vertical GO film is not feasible in the current system). Taken together, all these facts significantly lower the practical implications of the proposed method.

Responses to Reviewer Comments

Reviewer #1

Comment: The manuscript entitled “Controlling Nanochannel Orientation, Length, and Width in Graphene-Based Nanofluidic Membranes” describes the fabrication of a vertically aligned stacked graphene oxide membrane that is tested for gas separation.

This is a very well written paper that describes step-by-step the rationale of the process of making this vertically aligned GO membranes. The study was very comprehensive in addressing potential leaking issues, normalizing performance to active area and thickness, and describing change in performance over time. Therefore, this is a very complete work. It is also very novel. Horizontal stacked GO membranes have been used abundantly in the literature for different separation applications however this vertical design of the membrane has not been considered. The current manuscript covered well the literature on this topic, the limitations of horizontally stacked GO membranes, and how the vertical design address this issue.

I believe this paper can be of high interest to the field and inspire a new wave of GO-based membrane fabrication studies. However, to successfully to do, I encourage the authors to consider a few elements of discussion that would significantly help how to take this work to the next steps.

Choice of Zr: The manuscript is a bit light on why Zr was chosen as the intercalation ion. It mentions, at lines 89-91 “It was found that Zr doping using the water soluble $ZrOCl_2$ as an additive to the deposition suspension produced more adherent films and more regular wrinkle textures (Supplementary Fig. 1b and 2).” Why was Zr chosen for intercalation? The SI goes into more details about the need for having positive charge to maintain colloidal stability, which makes sense. However, they also show data with Fe and show that Fe leads to higher gaps, suggesting that Zr is better. Why? Were the experiment with Fe done at the same zeta potential? Would the same results be obtained with Cu? Or Ca? This is an important aspect and could

represent an important element to tune in the design. More discussion on the choice of the intercalation ions would be useful.

Response:

We appreciate the reviewer's overall positive response to the manuscript. Regarding the choice of ZrO(II), we went through a systematic selection process to pick the most effective cation, and this selection process and the scientific logic behind it is now documented in detail in this revised version. In essence, we wanted net positive charge on the metallized GO nanosheets to improve adhesion between the GO film and negatively charged substrate to prevent local delamination flaws or bubbles. We considered a wide variety of cations and screened their ability to create high positive zeta potentials on GO-surfaces through binding (see new Fig. S2b). We supported the screening experiments with a model of cation complexation and electrostatic charge screening for M^{n+} -GO suspensions (also new Fig. S2b and discussion). These included Cu^{2+} and Ca^{2+} that the reviewer suggested specifically. We picked the most promising two (Fe(III) and ZrO(II)) based on positive charge density and colloidal stability and fabricated films. Both ZrO(II) and Fe(III) improve film adhesion and improve the regularity of the wrinkled textures (Fig. S2a,c,d). ZnO(II) proved to produce better film structures than Fe(III), a trend that is consistent with the model predictions that show higher zeta potentials in ZnO(II)-GO suspensions (higher positive charge density), and we believe the model may find further use in the systematic design of other graphene/metal-cation co-suspensions. We agree with the reviewer that the scientific logic behind the ZrO(II) choice was not explained in enough detail in version 1. We have now added a new table of cation/carboxylate binding constants (Supplementary Table 1), a new figure panel (Supplementary Fig. 2b) and detailed discussion of the above points in the SI. We also summarized the discussion in the main article with this passage:

“We observed that neat (metal-free) wrinkled GO films contain voids at the interface of the film and bottom substrate, and the wrinkle deformation is irregular, producing chaotic microstructures seen in cross-section (Supplementary Fig. 2a). Increasing the mechanical strength of GO films (by increasing thickness from 1 to 2 μm) did not completely eliminate the distorted deformation or the voids. We hypothesized that increasing the substrate-film interfacial adhesion would prevent local delamination and eliminate the voids thus improving wrinkle

uniformity. Since both the GO nanosheets and the air-plasma-treated polymer surface are negatively charged, we explored metal cation addition as an electrostatic bridge between film and substrate. After considering a range of candidate cations through experiments and modelling (see Supplementary Table 1, Supplementary Fig. 2 and associated discussion), $ZrOCl_2$ was chosen as the optimal additive to the deposition suspension. The $ZnO(II)$ cation binds successfully to GO and reverses the zeta potential from -48 to +40 mV, and resulted in more adherent films and more regular wrinkle textures (Supplementary Fig. 2b-d). The colloidal behaviors of metal ion-GO suspensions were explored using a model of cation complexation and electrostatic charge screening, and the model predicts that the effective surface charge flipping of in Zr-GO suspensions was achieved by the very high stability constant of the $ZrO(II)$ -carboxylate complex (see detailed discussion in SI).”

The following passage was added to the SI:

“We attempted to achieve a more regular zig-zag pattern by creating stronger film/substrate adhesion. In our previous report¹, metal cation complexation proved to be effective in preparing colloidally stable, positively charged GO nanosheet suspensions. The binding of metal cations with fully charged acidic sites on GO can compensate for the negative charge, or even reverse the charge at sufficiently high metal-carbon ratio. As shown in Supplementary Fig. 2b, the pure GO aqueous suspension is colloidally stable due to (-/-) repulsion of ionized nanosheets, reflected by a zeta potential of -48 mV at $M^{n+}/C = 0$. Progressing addition of $ZrO(II)$ or $Fe(III)$ cations first reduces the magnitude of the negative charge then induces a charge flip and eventually a second regime of colloidal stability at high positive zeta potential ($> +40$ or $+20$ mV) for Zr/C or Fe/C ratios larger than $1/22$. In contrast, other metal cations were observed to cause flocculation at M^{n+}/C ratios greater than $\sim 1.5/1$, corresponding to the low-surface-charge window ($-15\text{mV} < \zeta < +15$ mV). The C/O atomic ratio of our GO is 2.1, therefore, the C atom molar concentration of our 0.1 mg mL^{-1} GO suspension can be estimated as $(0.1 \text{ mg mL}^{-1}) \times [2.1 / (2.1 \times 12 \text{ mg/mmol} + 1 \times 16 \text{ mg/mmol})] \cong 5 \text{ mM}$.

The experimental ζ -potential behaviors can be understood through a simplified model of cation complexation¹. The stability constants $\beta = [M^{n+} - GO^*] / [M^{n+}][GO^*]$ were estimated using cation-acetate complexes (Supplementary Table 1.), where GO^* represents an oxygen-containing binding site on GO with unit negative charge.

Supplementary Table 1.
Standard state stability constants for acetate complexes at 25 °C and 1 bar²⁻⁴

Metal	log β
ZrO²⁺	6.18
Fe³⁺	4.29
Al³⁺	3.02
Pb²⁺	2.70
Ni²⁺	2.12
Co²⁺	1.93
Cu²⁺	2.40
Ca²⁺	1.12

The theoretical surface charge density on Mⁿ⁺-GO nanosheets can be calculated by summation of ionized species on GO nanosheets as⁵

$$\sigma = \frac{z_i e c_i N_A}{A \rho} = \frac{e N_A (-1 \times GO_{acidic\ sites} + n_i \times [M^{n+} - GO^*])}{A \rho} \quad (1)$$

Where z_i is the valency of i th species, c_i is the concentration of bound i th species, $e = 1.6 \times 10^{-19}$ Coulombs, N_A is Avogadro's constant, A is theoretical specific surface area of GO (estimated as $1609 \text{ m}^2 \text{ g}^{-1}$ by combining theoretical surface area of graphene ($2630 \text{ m}^2 \text{ g}^{-1}$) and atomic C/O ratio of GO ~ 2.1), ρ is mass concentration of GO (0.1 mg mL^{-1}). n_i is effective charge of each Mⁿ⁺-GO binding site, and based on the zeta potential trends of Mⁿ⁺-GO suspensions, we proposed a multi-binding mode for Mⁿ⁺-GO nanosheets, where monoatomic cations (Fe(III), Al(III), Co(II), Ni(II), Pb(II), Cu(II) and Ca(II)) interact with two complexation sites on GO nanosheets, while diatomic cation that contains oxygen, like ZrO(II) interacts with only one complexation site. Therefore, n_i is 3/2 for Fe(III)-, Al(III)-; 2/2 for Pb(II)-, Co(II)-, Ni(II)-, Cu(II)-, Ca(II)- and 2/1 for ZrO(II)-GO colloids.

The Gouy-Chapman equation (2) was used to relate the surface charge of Mⁿ⁺-GO colloids with their zeta potentials, adopted describing a distribution of dissolved ions at a charged surface (electrical double layer)^{5,6},

$$\sigma_s = \frac{2\epsilon k T \kappa}{ze} \sinh\left(\frac{ze\zeta}{2kT}\right) \quad (2)$$

where z is the valency of the counter-ions, ε is the solution permittivity ($\varepsilon = \varepsilon_r \varepsilon_0$, $\varepsilon_r = 78.5$ is obtained on DLS), k the Boltzman constant, T the temperature and κ the reciprocal of Debye length (nm^{-1}). The Debye length is given by the expression $0.304/\sqrt{I}$, where I is the ionic strength defined as $\frac{1}{2} \sum z_i^2 [x_i]$; x_i is the molar concentration of the i th species, and z_i is valency. All M^{n+} -GO colloids were prepared in 20 mM NaNO_3 aqueous solutions to minimize the effect of different ionic strength on zeta potential (as the conc. of test salts range from 0.05 – 15 mM).

By combining equation (1) and (2), the concentration of total acidic sites on GO nanosheets was back-calculated as ~ 0.03 mM (zeta potential of 0.1 mg mL^{-1} GO suspensions is -48mV). Similarly, by combining equation (1), (2) and the binding stability equilibrium relations, theoretical zeta potentials of M^{n+} -GO were back-calculated at different metal-carbon ratios⁵. As shown in Supplementary Fig. 2b, the theoretical and experimental zeta potentials are in good agreement, indicating the successful surface charge reversion of ZrO(II) and Fe(III) is a synergetic effect of high complexation constant and effective valence as in the proposed model. The (+/+) repulsion between Zr- or Fe-GO nanosheets maintains colloidal stability and produces high quality films during subsequent drying induced assembly.

Multivalent metal cations ZrO(II) and Fe(III) were doped into the films by depositing GO from ZrOCl_2 or $\text{Fe}(\text{NO}_3)_3$ solutions to introduce (+/-) electrostatic attraction at the GO-substrate interface. The side views of Zr- and Fe-GO wrinkled films are shown in Supplementary Fig. 2c-d. Fe- and Zr-doped GO wrinkled films possess a more regular zig-zag pattern. Furthermore, the Zr-GO wrinkled films (Supplementary Fig. 2c) exhibit higher ridge curvature (sharper corners) and stronger affinity (no detachment voids) than the Fe-GO films (Supplementary Fig. 2d), which may be due to the higher zeta potential on Zr- (+40 mV) than Fe-GO (+20 mV) nanosheets at same metal-carbon loading ($M^{n+}/C = 1/22$), which maximizes the doping effect and facilitates the subsequent sectioning. The further development of VAGMEs was therefore pursued using Zr-doped GO.

Supplementary Figure 2. Morphologies of wrinkled films and ζ -potential behaviors of metal cation-GO nanosheet suspensions. **a**, Side (cross-sectional) views of epoxy-fixed neat GO wrinkled films. Left, 1 μm -thick GO films after compressive deformation and embedding in epoxy. Scale bar, 20 μm . Right, 2 μm -thick GO films after compressive deformation and embedding in epoxy. Scale bar, 50 μm . **b**, Experimental and theoretical (model-predicted) ζ -potentials of GO nanosheet suspensions with varying degrees of metal salt addition, expressed as a function of M^{n+}/C atomic ratio. A series of metal cations are tested, including ZrO(II), Fe(III), Al(III), Ni(II), Co(II), Cu(II) and Ca(II). GO dispersions are all at 0.1 mg mL^{-1} solids loading. Colloidal stability of ZrOCl₂ doped GO can be achieved by $\zeta > +20\text{mV}$ (+/+ repulsion due to surface charge inversion). **c**, Sides views of metal ion-doped GO/epoxy composites. 1 μm -thick Zr-GO films after compressive deformation and embedding in epoxy. **d**, Sides views of metal ion-doped GO/epoxy composites. 1 μm -thick Fe-GO films after compressive deformation and embedding in epoxy. Scale bar, 20 μm .”

Comment: Thickness and active area: The study measure water vapor flux and normalized the flux to thickness and active area to show that the vertically aligned 1 GO membranes had higher permeability than horizontally stacked GO membranes. For real life considerations, it is the permeance that matters though, and not the permeability. However, higher permeability can be very important IF there is a path to increasing the permeance, by making the membrane thinner or increasing the active surface. The SI discuss how the sample thickness was a major challenge

in obtaining clean cuts without cracks and defects. Will this be a limitation that will prevent ever reaching permeance values that are of interest for real life applications? I suggest to include a paragraph of discussion of the potential avenues the authors see to improve the performance of the membrane and what are the challenges to do so.

Response:

We agree it is a current challenge to prepare vertical GO membranes with very high active area fraction. This is the first study of this concept, however, and as such it is focused on demonstrating how compressive wrinkling can be used to create near vertical alignment. Even this first study achieves an increase in active area by a factor of 300 relative to the simple tilting of planar GO films. Looking ahead, we believe there is a significant opportunity to improve and optimize the wrinkling approach. Follow-on studies will likely pursue higher compression ratios (higher tilt angles) because there is no known fundamental limitation on improvement by this method, and the active area fraction can theoretically approach unity (the tightly folded “all-edge” membrane) as tilt angle approaches 90 degrees. Related wrinkling methods may be explored that utilize stamping or grating templates. Other opportunities include improving the interfacial interaction between GO and the embedding matrix, and exploring better de-capping methods (including a variety of chemical etching approaches as alternatives to mechanical sectioning). We have added a brief discussion near the end of the main text.

Relevant additions to the main text:

In the Introduction section.....

“Another method to obtain vertical GO nanochannels is to tilt an entire planar GO film by 90 degrees. Such simple tilting, for a 1 μm thick GO film, produces only a single 1 μm wide slit rather than a membrane, and hand stacking of multiple micrometer-slits has been used to create novel devices for study of nanopore transport and molecular sieving¹⁴. Those devices are not thin membranes, but rather nanochannel-embedded blocks with diffusion path lengths in the mm size range that are primarily useful for scientific experiments.”

”

In the Results section.....

“Overall, the key concept in this study is to geometrically deform a planar film, in a single processing step, to produce hundreds of tilted short film segments, each segment representing an array of ~1000 near-vertical nanochannels for rapid, direct (non-tortuous), through-membrane transport. The technique achieves a 300-fold increase in active area (face area occupied by the entrances to nanochannel arrays) relative to the simple hand tilting of a complete planar GO film. The current proof-of-principle demonstration uses a tilt angle of 74 degrees to produce an active area fraction of 10%, but this may be improved through further work at higher compression ratios (higher tilt angles) possibly using cycled contraction³¹. There is no known fundamental limitation on improvement by this method, and the active area fraction theoretically approaches unity as tilt angle approaches 90 degrees (the ideal tightly folded “all-edge” membrane). Improvements may also be expected through alternative wrinkling approaches based on stamping or grating templates, or new de-capping methods that include chemical etching as alternatives to mechanical sectioning³².”

Comment: Table 1 – error bars need to be added to the VAGME results.

Our response: The vapor fluxes were plotted with error bars in Fig.3 but without detailed error bars in Table 1 -- we apologize for this omission. Table 1 was updated in the revised manuscript with error bars.

“Table 1 Water vapor fluxes measured through VAGME devices and conventional GO films

Membrane	Flux (60 °C) kg m ⁻² hr ⁻¹	Flux (80 °C) kg m ⁻² hr ⁻¹	Flux (100 °C)** kg m ⁻² hr ⁻¹
Conventional GO films	1.7 - 4.1* ^{9,27}	(film cracking)	(film cracking)
VAGME (total area basis)	1.9±0.1	5.6±0.2	13±2
VAGME (active area basis)	27±1	79±2	(1.8±0.3)×10 ²
	Normalized flux (60 °C) kg m ⁻² hr ⁻¹ μm	Normalized flux (80 °C) kg m ⁻² hr ⁻¹ μm	Normalized flux (100 °C) kg m ⁻² hr ⁻¹ μm
VAGME (active area basis)	(5.4±0.2)×10 ²	(1.6±0.1)×10 ³	(3.5±0.6)×10 ³

* The flux value from ref. 9 includes a calculated temperature correction based on vapor pressure driving force.

** The 100 °C data is time-dependent and the table entries are average fluxes over 12 hrs.”

Comment:

GO synthesis: The paper says “GO nanosheets were prepared by a modified Hummers method”. However, modified Hummer’s method varies a bit between papers in the ratios of reagents to graphite, temperature, reaction time, etc. The authors cite a paper that is from 2017. However, that paper says: “Graphite oxide (GO) was prepared by applying a modified Hummers method [21, 44]” without more information on the actual quantities and ratios of the material, taking us down the rabbit hole of hunting for the actual experimental details (now with two references to look for to try to figure out what was done). This is not acceptable – the manuscript should have all the experimental details needed to be able to reproduce the experiments as closely as possible. This can go into the SI.

Response:

The fabrication process of GO can be essential for reproducing experiments, and we apologize for omitting the details in our previous version. We have now included a full protocol in the SI of revision:

“Synthesis of GO suspensions

Graphite oxide (GO) was synthesized using a modified Hummers’ method with a pre-intercalation treatment. Concentrated H₂SO₄ (100 mL) was heated to 80 °C in a 500 mL Erlenmeyer flask. K₂S₂O₈ (10 g) and P₂O₅ (10 g) were added to the acid and stirred until fully dissolved. Graphite powder (14 g, Bay Carbon Inc. SP-1 grade) was added to the solution and kept at 80 °C for 5 hrs. The mixture was then cooled and placed in an iced bath, followed with slow dilution by 200 mL deionized (DI) water. The solid sample was filtered using a vacuum filtration unit (47 mm diameter, 0.2 μm pore size), further rinsed with 1 L DI water and then dried in air for overnight. Concentrated H₂SO₄ (500 mL), NaNO₃ (10 g) and the pre-oxidized

graphite powder were transferred into a 2 L Erlenmeyer flask in an ice bath, then KMnO_4 (70 g) was slowly added to the mixture while stirring (temperature was controlled within $10\text{ }^\circ\text{C}$ during the complete process). The flask was moved to a $40\text{ }^\circ\text{C}$ water bath and left for 3 hrs, and then transferred back into the ice bath. DI water (1 L) was slowly added to the flask while stirring, taking caution that the temperature did not rise above $55\text{ }^\circ\text{C}$. After dilution, 60 mL of a 30% H_2O_2 solution was added to the mixture dropwise. The color of the solution turned from dark brown to bright yellow. This mixture was left under stirring overnight to consume the H_2O_2 . The washing process included 5 rounds of acid washing using 1M HCl to remove residual salts (centrifugation at 4000 rpm for 30 min), and 4 rounds of acetone washing (centrifugation at 4000 for 30 min). After thorough washing, the resulting wet solid was collected and dried in air for 72 hrs. To harvest GO suspensions, ~ 4 g dried sample was fully dispersed in 1 L DI water and bath sonicated for 40 min. The suspension was centrifuged at slow speed (around 1000 rpm) for 5 min, then the supernatant was carefully collected by a pipette. After 3 rounds of centrifugation and supernatant collection, the final GO suspension was obtained for usage. The as-prepared GO nanosheets possess a lateral size $\sim 1\text{ }\mu\text{m}$ and thickness $\sim 1\text{ nm}$ (Supplementary Fig. 1a). Potential GO impurities N, S, Mn, K, Cl, and P were not detected by XPS. The C1s XPS spectra for GO is shown in Supplementary Fig. 1b, the C/O atomic ratio of GO is ~ 2.1 .

Supplementary Figure 1. Morphology and composition of as-prepared GO nanosheets. a, AFM image and accompanying height profile of GO nanosheets drop-cast from diluted suspension onto mica. **b,** C1s XPS spectra of a GO film.”

Reviewer #2

Comment:

In this manuscript, Liu et al. reported the use of a self-assembly method to fabricate vertically aligned graphene membranes. Briefly, GO films are first cast onto a pre-stretched polystyrene substrate, which was then compressed in a horizontal direction to create unidirectional periodic wrinkle patterns in vertical direction. The wrinkled GO films were removed from the substrate and imbedded into epoxy resin, which could be sectioned into thin, composite GO/epoxy membranes. The fabricated membranes were characterized on SEM to verify the aligned structure of GO. Additionally, the authors demonstrate the molecular selectivity of membrane on a vapor permeation setup using water or hexane as feed water.

Generally, the method using epoxy embedding, followed by microtoming to create vertical GO film is not novel, because a similar method has been reported in a previous study (Nair et al, Nature Nanotechnology, 2017, DOI: 10.1038/nnano.2017.21). Such fact significantly lowers the novelty of this study. Additionally, the lack of systematic investigation and practical implications decrease the overall quality of this work, and some key conclusions in the manuscript are also not sound. Detailed comments are listed below.

Comment 1 In Figure 1, the authors illustrate the overall procedures and ideas on fabricating vertical GO film. As I mentioned above, it is not clear how their procedures are different from the previous study using the similar epoxy technique. They used a pre-stretched polystyrene substrate to create wrinkled GO films (Figure 1b). Such procedure is also not necessary for obtaining the vertical GO structure. For example, the epoxy embedding (figure 1c) could be directly applied to the deposited GO film. After that, the vertical alignment could be easily achieved by manually rotating the angle of the film in microtoming (Figure 1d).

Response:

We agree that the Nair et al. 2017 study is relevant here and was in fact cited in our original manuscript (reference 14). Specifically, we used the Nair et al. work to support our choice of

water-impermeable epoxy. The two studies, however, are very different both in the overall objective and the assembly concepts.

The goal of the Nair et al. (2017) paper was to control the interlayer spacing of GO nanochannels by suppressing swelling and then test the corresponding effects on liquid phase molecular sieving behavior. Epoxy was employed as physical confinement and glue between GO strips. The Nair et al. paper does not produce usable thin membranes with large face area, but rather a hand-fabricated block with imbedded nanochannels as a convenient device for performing fundamental scientific experiments on nanochannel transport. The transport pathlengths through that block are 3 mm long (vastly greater than the active selective regions in molecular sieve membranes). This was a high quality paper, but reducing diffusion pathways in membranes and aligning nanosheets by self-assembly were not their objectives.

Secondly, Nair et al. manually tilted the GO planar film followed with epoxy resin embedding. This process produces a micrometer-wide slit instead of a typical membrane with macroscopic face area. Therefore, Nair et al. manually stacked and glued GO slits for multiple cycles to obtain a laminated GO/epoxy rectangle, followed by incorporation of multiple rectangles into a stainless-steel plate for the pressure filtration experiment. In contrast, our study uses compressive self-assembly to efficiently transform one single GO planar film into 300 near-vertical segments in one step, each segment comprising about ~1000 nanochannels. This self-assembly method achieves active areas (containing nanocapillaries) that are 300 times larger than the manual tilting of a GO planar film, and our final product is a 4 mm in diameter, 20 μm thick, sheet-like membrane that is a significant step toward practical membrane devices.

Finally, unlike the Nair paper, the present work offers a systematic study of vapor permeation at elevated temperatures/pressures, where the VAGME exhibits very high flux of water and selectivity.

Except for the use of epoxy resin embedding, and our minor conclusion about swelling inhibition (which confirms the Nair et al. conclusion), the goals and methods in the two studies are quite different. To help clarify this in the manuscript, we added these several brief discussion passages into the main text:

in the Introduction.....

“Another method to obtain vertical GO nanochannels is to tilt an entire planar GO film by 90 degrees. Such simple tilting, for a 1 μm thick GO film, produces only a single 1 μm wide slit rather than a membrane, and hand stacking of multiple micrometer-slits has been used to create novel devices for study of nanopore transport and molecular sieving¹⁴. Those devices are not thin membranes, but rather nanochannel-embedded blocks with diffusion path lengths in the mm size range that are primarily useful for scientific experiments.”

in the Results section.....

“Overall, the key concept in this study is to geometrically deform a planar film, in a single processing step, to produce hundreds of tilted short film segments, each segment representing an array of ~ 1000 near-vertical nanochannels for rapid, direct (non-tortuous), through-membrane transport. The technique accomplishes a 300-fold increase in active area (face area occupied by the entrances to nanochannel arrays) relative to the simple hand tilting of a complete planar GO film. The current proof-of-principle demonstration uses a tilt angle of 74 degrees to produce an active area fraction of 10%, but this may be improved through further work at higher compression ratios (higher tilt angles) possibly using cycled contraction³¹. There is no known fundamental limitation on improvement by this method, and the active area fraction theoretically approaches unity as tilt angle approaches 90 degrees (the ideal tightly folded “all-edge” membrane). Improvements may also be expected through alternative wrinkling approaches based on stamping or grating templates, or new de-capping methods that include chemical etching as alternatives to mechanical sectioning³².”

Comment: 2. According to Figure 1e, the authors considered their GO nanosheets with a uniform size (or at least small size distribution). However, most of the GO suspension has a much larger size distribution. As each channel is formed by two laboring nanosheets, the selective channel length is likely determined by the shorter GO sheets. Microtoming of the composite film might only expose the longer GO sheets. In this case, the majority of selective channels were still embedded in the resin, which could not exhibit any permeability.

Response: The reviewer is correct that our GO nanosheet samples have a significant distribution in lateral dimension. We believe, however, there is a slight misunderstanding regarding how this

distribution impacts structure and behavior due to our lack of clarity in the assembly description. We have refined Fig. 1 to better illustrate the process. As shown in Fig. 1e, the brown strips are not single sheets, but rather multilayer **GO film segments** created by tilting and sectioning. Each has a dimension of ~ 3 mm (length, into the image plane) $\times 1$ μm (width) $\times 20$ μm (height). Each GO film segment consists of ~ 1000 layers of **GO nanosheets** (as depicted in the inset of 1e) creating about 1000 nanochannels. From the refined Fig. 1e we can see the size distribution of GO nanosheets and the implications of that distribution. The smaller sheets are not cut off, or made inactive, but exist within the layered film segment and continue to contribute to permeation and selectivity. Also, an AFM image of our GO suspension was added into SI, the as-prepared GO nanosheets possess a lateral size ~ 1 μm , thickness ~ 1 nm (Supplementary Fig. 1a).

We also modified the caption of Fig. 1 to emphasize that brown strips represent 1 μm thick multilayer GO film segments and the light-yellow box represents epoxy matrix. As mentioned in the main text, the epoxy resin is used as embedding media and cannot diffuse into and block the GO nanochannels. Therefore, once the sharp connecting corners of GO microfilms are removed by microtome sectioning in Fig. 1d, the ~ 1000 nanochannels in each GO strip segment are exposed and fully open (as shown in modified Fig. 2b and Supplementary Fig. 4b).

Figure 1. Structure and fabrication steps leading to vertically aligned Zr-GO/epoxy membranes. i. Sketch of compressive film wrinkling. Side view of planar and 1D wrinkled films. Figure illustrates that wrinkling tilts the horizontal line into multiple near vertical line segments. **ii. Fabrication steps leading to vertically aligned Zr-GO/epoxy membranes.** **a**, Drying-induced assembly of Zr-GO nanosheets (Zr/C atomic ratio $\sim 1/22$) on pre-stretched polystyrene substrate (GO film thickness $1\ \mu\text{m}$). Inset: nanostructure of planar Zr-GO films with horizontal alignment and tortuous flow pathways. Brown strips represent $1\ \mu\text{m}$ thick multilayer GO film segments. The light-yellow box represents the epoxy matrix. Purple spheres represent ZrO(II) cations (unhydrated state for reference). Fully hydrated ZrO(II) diameter is $\sim 1\ \text{nm}$, implying that ZrO(II) likely exists in the interlayer spaces in a partially hydrated state complexed with O-containing groups on GO. **b**, Wrinkled Zr-GO films are produced by thermally activated mechanical compression. **c**, Wrinkled Zr-GO films are removed from the substrate and imbedded into epoxy resin. **d**, Multiple cycles of microtome sectioning yields Zr-GO/epoxy composite membranes. **e**, Side view of vertically aligned Zr-GO/epoxy membrane (VAGME) with the entrances to

interlayer nanochannels open at the top and bottom surface. This method transforms a single planar Zr-GO film into hundreds of vertical film segments, where each segment is an array of ~ 1000 GO interlayer nanochannels aligned with a strong Z-directional component.

Supplementary Figure 1. Morphology and composition of as-prepared GO nanosheets. **a**, AFM image and accompanying height profile of GO nanosheets drop-cast from diluted suspension onto mica. **b**, C1s XPS spectra of a GO film.

Comment: 3. The authors only provided some SEM micrographs (Fig. 2) to sustain their claim on the formed vertical GO structure. However, these rough images are good evidence to verify the nanosheet alignment. More importantly, through SEM observation, I can see that most of GO nanosheets were aligned with some tilted angles, which is far from the authors' definition of "vertical alignment". Therefore, more systematic material characterization is needed to verify the alignment of GO.

Response:

We appreciate this comment and have reworded our "vertical alignment" claim. It is true that the nanochannels in the current device are not strictly vertical, but all have finite tilt angles. As mentioned in the main text, "the unidirectional relaxation ratio of $3/11 = 0.272$ corresponding to $\theta = 74$ degree rise angles in the ideal zig-zag geometric model (Fig. 1 part i). Observed rise

angles are in the range 60-80 degrees (Fig. 2)". The goal of realignment was to create "direct passage nanochannels" that do not require permeating species to navigate tortuous pathways across the film plan and thus avoid the $\sim 10^3$ tortuosity factor in conventional membranes. This goal is actually achieved as soon as the tilt angle is sufficient to allow the de-capping process, and any further tilting toward 90 degrees further reduces the "direct passage" distances, but only modestly. For example, for a given membrane thickness, if the 90 degrees pathlength is l , then the pathlength achieved by tilting to only 74 degrees is $1.04l$ (a 4% difference). Thus the near-vertical channels achieve almost all of the benefit of truly vertical channels, and the first version of the manuscript used "vertical nanochannels" as a simple descriptor of the structure. We do wish to make accurate statements and claims, however, so are happy to take the advice of the reviewer and modify the term "vertical" to "near-vertical" in the text (many places).

Comment: 4. The author claimed that "microscopic inspection of our final VAGME membranes did not reveal visible flaws" (line 163). It sounds like the authors considered the SEM morphology as the first evidence to demonstrate the integrity of their films. However, it should be noted that their SEM micrographs only allow the observation of structure on the micrometer scale (evidenced by the scale bars in the figures), which is at least three orders of magnitude larger than the selective GO-channel size (less than 1 nm). In other words, even some defects (<100 nm) were present in the film, it is challenging to observe on their SEM micrographs. But these defects were detrimental to the selectivity of the composite film. Therefore, in-depth material characterization is also needed to reveal the actual assembly behaviors of GO nanosheets.

Response: We agree fully with the reviewer's scientific point that SEM will not reliably reveal sub-100nm defects, and that these may be enough to compromise membrane selectivity. Our SEM inspection, however, is just a first step in a multistep process to assess membrane flaws. The SEM is designed to reveal micro-scale flaws, which may be introduced in the form of (i) trapped air bubbles in the epoxy, which become holes in thin sections, (ii) cracks in the GO films, and (iii) delamination openings at the epoxy-GO interfaces occurring during curing or processing. Screening for microscale defects then qualifies a membrane for further examination in a second step. We propose that defects down to ~ 1 nm cannot be reliably ruled out through imaging, no

matter how detailed and exhaustive, so instead we used a different technique: selective molecular permeation. Many aspects of the molecular selectivity of GO nanochannels are well known in the literature, such as the ability to pass water but not hexane, so this selectivity was used here as a test for the absence of membrane defects. Hexane is a small molecule, and even $< 10\text{nm}$ flaws in the membrane would pass hexane readily, and destroy the water/hexane selectivity that is characteristic of GO nanochannel transport. We believe the purpose of the water/hexane experiments in the manuscript was not fully clear - it was not meant to provide new information on the selective permeation properties of GO nanochannels (which are already well known), but as a test for membrane defects not easily visible by SEM.

We believe the original section in the text was not sufficiently clear on this point, and we have revised it to this: “Microscopic inspection of our final VAGME membranes did not reveal visible micro-scale defects. This inspection technique is not reliable for identifying defects $< 100\text{ nm}$, however, which can also degrade membrane selectivity. We therefore used selective molecular vapor permeation as a probe for the presence or absence of nanoscale defects. A distinctive behavior of GO interlayer nanochannels is their ability to pass water vapor rapidly, while excluding all non-polar and many other polar molecules^{3,9,26}, including hexane²⁷. In the absence of membrane defects, transport will occur only through nanochannels with very high permeation rates for water vapor and near perfect exclusion of hexane vapor. In contrast, defects significantly larger than the sub-nm interlayer channels would pass hexane and water at similar rates (for similar vapor pressures) governed by their similar gas-phase diffusion coefficients. Therefore, single-phase permeation experiments with water and hexane vapor were carried out to test the hypothesis that transport in these membranes occurs primarily through GO interlayer nanochannels rather than defects.”

Regarding in-depth characterization of GO nanosheet assembly, we would like to point out that the assembly and structure of GO films or papers has been extensively studied in the literature, and our films are typical of those fabricated and studied by many others. We added compressive wrinkling, but this introduces structural changes primarily on the microscale, which have also been extensively studied by ourselves and others. We do now include a time-resolved

XRD experiment that tracks the appearance of lamellar structures in Zr-GO films during drying, as well as a top view of the surface of VAGME showing the alignment of nanosheets in SI.

Content added in main text:

“The high magnification view (Fig. 2b and Supplementary Fig. 4) of an individual Zr-GO strip shows a tightly packed, multilayered structure, which appears identical to original planar Zr-GO films. The assembly and alignment of Zr-GO nanosheets in original Zr-GO films can be found in SI.”

Contents added in SI:

“Casting Zr-GO nanosheet suspension onto substrate produces intercalated laminates as shown by time-resolved XRD in Supplementary Fig. 4a. During the drying process, the Zr-GO nanosheets slowly stack together and form a lamellar structure, exhibiting an interlayer spacing of 8.8 angstrom. After realignment into VAGME, the GO strips still exhibit multi-layered structures as shown in Supplementary Fig. 4b.”

Supplementary Figure 4. Assembly of GO nanosheets. **a**, Time-resolved XRD tracking the appearance of lamellar structures in Zr-GO films during drying. **b**, Top view of Zr-GO strips on the VAGME surface, exhibiting multi-layered structure.

Comment: 5. Regarding the vapor permeation experiments, the authors used this method to demonstrate the integrity and selectivity of the GO membrane. There are a couple of defects in their experimental design.

a. First, membrane separation of water and hexane is not needed as they would form two phases in nature. The authors need to provide some background explanation on selecting these two solvents as feed water.

Response:

This is a reasonable comment. Indeed, we did not do these experiments to demonstrate a method for water / hexane separation, which would be trivial (one could use a decanter or stripping). We used them as known reference materials in an assay to probe for membrane defects (described above and the next passage for the revised manuscript is given above).

Comment: b. Second, even if the authors aimed to utilize the size difference in these two solvents to demonstrate the selectivity of their fabricated GO films, distinct differences in the physicochemical properties of two solvents would limit mechanistic elucidation. For example, physiochemical properties such as hydrophilicity/hydrophobicity and surface tension may contribute to the transport of these vapor in the composite film, which in turn affect the selectivity. **The authors need to select more solvents as vapor feed for a comprehensive evaluation of membrane selectivity**, thereby excluding the possible contribution of other properties. Additionally, the authors performed the vapor permeation experiments using feed with a single solvent (either water or hexane). **At least mixed vapor of water and hexane should be used to verify their claim on the selectivity.**

Response:

The reviewer provides an accurate summary of our selected molecules. As a first comment, it was never our goal to systematically characterize selectivity, since many prior papers have done this already for GO nanochannels. Our goal was to create a new self-assembled membrane structure, and use permeation as a probe to prove that transport through the membrane was restricted to the nanochannel arrays (not defects). Of course we understand the interest in more in-depth characterization of selectivity, and we thus took the reviewer's suggestion and added 2-propanol to our test matrix, which is a hydrophilic but larger molecule (0.49 nm) with a three-dimensional structure (unlike hexane). Very low permeation results from the water-soluble 2-propanol is further evidence for nanochannel-like size exclusion of VAGME. We also accepted

the reviewer's request to measure fluxes of water and 2-propanol from their liquid mixtures, which also shows high selectivity for water over 2-propanol (including appropriate corrections for the equilibrium vapor pressures above the non-ideal liquid mixtures). We have included results and discussion in the paper.

Content added to the main text:

“Fig. 3 shows the test apparatus²⁷ which is heated to produce a range of vapor pressures up to 1 bar corresponding to the normal boiling points of water (100 °C), hexane (68 °C) and 2-propanol (82 °C). Magnetic stirring is introduced to reduce mass transfer resistance on the upstream side, and the device was operated in a fume hood to provide convective flow over the VAGME surface to reduce downstream mass transfer resistance. Water vapor transmission rates through VAGME films show a linear dependence on upstream water vapor pressure and approach 10 mmol mm⁻¹ hr⁻¹ at 1 bar vapor pressure, while (pure) hexane or 2-propanol permeation is at or below the detection limit (Fig. 3b). Pure epoxy films (20 μm thick), used here as a negative control sample, show no measurable flux for all three solvent vapors indicating flaw-free full density membranes (Fig. 3c).

The vapor transport behavior associated with water/2-propanol liquid mixtures is shown in Fig. 3b. In this liquid mixture vapor, VAGME allows water permeation but excludes 2-propanol molecules. As shown in Supplementary Table 2 and Fig. 3b, the flux of water or 2-propanol in a mixture is comparable with that of individual vapor. In general, water molecules enter nanochannels driving by capillary force, and favor to flow through non-oxidized region^{3,12}. The free spacing of Zr-GO nanochannels can be calculated as 0.35 nm after deduction of carbon grid and functional groups (0.20 nm)^{15,28}, which is capable to exclude hydrophilic 2-propanol molecules with a kinetic diameter of 0.49 nm²⁹. The rejection of hexane is mainly contributed by its hydrophobicity (water insolubility)⁹. The data clearly show that VAGME films exhibit the characteristic high selectivity of GO nanochannels for water over hexane or 2-propanol, and confirm the absence of pores and interfacial cracks that would degrade this selectivity.

Figure 3. Measurements of selective molecular transport through VAGME nanochannels. a, Custom diffusion cell for temperature-dependent vapor permeation experiments. VAGME films are adhered on the top of a copper sheet spanning a 4 mm diameter hole by epoxy adhesive, and the plate is clamped and sealed on top of a stirred liquid/vapor cell. **b,** Measured vapor fluxes through VAGME films for pure liquids and liquid mixtures. Pure liquids: water, n-hexane and 2-propanol. Liquid mixtures (circled in red dot line): water and 2-propanol. **c,** Control experiments for water, n-hexane or 2-propanol vapor permeation through epoxy-only membranes of same (20 μm) thickness. Flux values are normalized by the active Zr-GO (nanochannel array) area.”

Contents added in SI:

“Supplementary Table 2 shows the fluxes of water and 2-propanol vapor above their liquid mixtures. The water/2-propanol system is highly non-ideal and azeotropic, but vapor/liquid equilibrium data are widely available, and the component vapor pressures (partial pressures)

above the liquid mixtures are also given in Supplementary Table 2⁷. The VAGME selectivity for water over 2-propanol is above a factor of 100 times. The corresponding calculation process is shown in Supplementary Fig. 5 and Table 3.

Supplementary Table 2 Water and 2-Propanol vapor fluxes of liquid mixtures measured through VAGME devices.

Component in the feed liquid mixtures	Flux (80 °C) mmol mm ⁻² hr ⁻¹	Vapor pressure above non-ideal liquid mixtures* bar
Water	5.0±0.2	0.45
2-Propanol	(6.2±3.7) ×10 ⁻²	0.55

*Vapor pressures are estimated from vapor-liquid equilibrium plot for water-2-propanol⁷.

Supplementary Figure 5. Calibration curve of refractive index (n_D) and 2-propanol molar ratio in the liquid mixtures.

Supplementary Table 3. Calculation parameters for water/2-propanol liquid mixtures.

Mass loss of liquid mixtures (mg hr ⁻¹)	76.2±7.5
n_D (before)	1.3649±0.001
IPA mol% (before)	19.1±0.1%
n_D (after)	1.3654±0.001
IPA mol% (after)	19.4±0.1%

Comment: c. Fourth, it is not clear how these vapors would transport through GO channels. Will they interact with GO nanosheets while diffusing through GO nanochannels?

Response: The transport and exclusion of molecules in nanochannels has been widely studied in the literature. Here we have added a brief discussion in the main text.

Contents added:

“In general, water molecules enter nanochannels driven by capillary forces, and flow preferentially through non-oxidized regions^{3,12}. The free spacing of Zr-GO nanochannels can be calculated as 0.35 nm after deduction of carbon grid and functional groups (0.20 nm)^{15,28}, which is capable to exclude hydrophilic 2-propanol molecules with a kinetic diameter of 0.49 nm²⁹. The rejection of hexane occurs primarily through its hydrophobicity (water insolubility)⁹.”

Comment: d. Lastly, I am curious is there is any wetting of the GO film during the vapor permeance experiments. As GO is hydrophilic, it is likely that the water vapor condenses on the inner side of the film and gradually transport to the outer layer. In contrast, hexane vapor was repelled by the hydrated GO film, which may responsible for the observed non-flux of hexane in Figure 3b.

Response: Yes, we believe the rejection of hexane is primarily due to its hydrophobicity and this would prevent its liquid-like transport across the membrane, but could also suppress its adsorption on the surface to form a liquid-like film on the inner face. We have added related contents into the main text, as mentioned in previous response.

Comment: 6. Instead of performing vapor permeation experiments, **liquid filtration tests** using different solute-solvent pairs should be a more facile way to demonstrate the stability and selectivity of their fabricated films, particularly considering their envision on the potential application of the films as “membranes for ultrafiltration and reverse osmosis” (Line 238).

Response: We appreciate reviewer’s suggestion and conducted a liquid filtration test using NaCl and MgCl₂ salt solutions. Results and discussion on liquid filtration tests are added into main text and SI.

Contents added in the main text:

“Beyond vapor permeation, liquid filtration tests on NaCl and MgCl₂ aqueous solutions were also conducted and discussed. VAGME films show the expected high selectivity for Na(I) over the larger Mg(II) that is characteristic of GO nanochannel transport. More detailed discussion is included in the SI.”

Content added in the SI:

“All liquid filtration tests were done using the device shown in Supplementary Fig. 8. Permeation data for NaCl and MgCl₂ aqueous solutions through VAGME are shown in Supplementary Fig. 9 and Supplementary Table 4. The permeation rate of NaCl is two orders of magnitude higher than that of MgCl₂, which is consistent with the hydrated diameter difference between Na(I) and Mg(II)¹².

Supplementary Figure 8. Liquid filtration device. Experimental set-up showing H-shaped glass cells as feed and permeate compartments. VAGME was clamped between two O-rings and then fixed between feed and permeate compartments to provide a leak tight environment for the liquid filtration experiments.

Supplementary Figure 9. Salts permeation through VAGME. Permeation through a VAGME membrane from the feed compartment with 0.05 M aqueous solution of NaCl or MgCl₂.

Supplementary Table 4. Permeation rates of ions through VAGME and the hydrated diameter of metal ions.

Ions	Permeation rate mol m ⁻² hr ⁻¹	Hydrated diameter nm
Na(I)	1.1±0.1	0.45
Mg(II)	(1.9±0.2) × 10 ⁻²	0.80

”

Comment: 7. According to the data in Table 1, the benefit of vertical GO in gaining water flux enhancement is negligible. For example, the vertical GO film exhibited comparable water flux to that of the conventional GO film (1.9 vs 1.7~4.1 kg m⁻² h⁻¹). This result implies that the decrease of tortuosity, which resulted from the vertical alignment, is not that significant compared to the non-aligned GO film.

Response:

The reviewer is correct that the performance looks similar, but only when the comparison is made on a very particular basis: using total (not active) area on the membrane face and ignoring differences in membrane thickness. This basis is not the correct one to use to make conclusions

about tortuosity effects. The VAGME films are thicker (due to out of plane wrinkling) and have only 10% active nanochannel area (which can be improved through further development). To account for these differences, Table 1 also presents flux values properly normalized by nanochannel active area and thickness, and these exhibit a 300x enhancement over conventional membranes, which would be the best estimate of the reduced tortuosity effect (300x). Although the overall permeance of our current product does not yet reach its full potential due to limitations in active area and thickness, there is much room for improvement and optimization, and overall the compressive assembly concept is very promising for fabrication of advanced membranes. We discuss the potential to further develop and improve these VAGME features in the revised version (see below).

Comment: Even the authors argued to use effective areas for calculation, the water flux of the vertical film was enhanced to $27.1 \text{ kg m}^{-2} \text{ h}^{-1}$. Such water flux is still only around 6 times larger than that of the conventional film. More importantly, using effective area calculation does not sound as the epoxy is always needed for fabricating these vertical film (I would assume making epoxy-free vertical GO film is not feasible in the current system). Taken together, all these facts significantly lower the practical implications of the proposed method.

Response:

We acknowledge there is more work to be done to produce fully functional membranes that reach their potential for high performance. We are convinced, however, that this first paper demonstrates a new material architecture and self-assembly concept that is a significant step toward revolutionary improvement in GO membranes. Looking ahead, we believe there is a significant opportunity to improve and optimize the wrinkling approach. Follow-on studies will likely pursue higher compression ratios (higher tilt angles) because there is no known fundamental limitation on improvement by this method, and the active area fraction can theoretically approach unity (the tightly folded “all-edge” membrane) as tilt angle approaches 90 degrees. Related wrinkling methods may be explored that utilize stamping or grating templates. Other opportunities include improving the interfacial interaction between GO and the embedding matrix, and exploring better de-capping methods (including a variety of chemical etching

approaches as alternatives to mechanical sectioning). We have added a brief discussion near the end of the main text (as described above).

REVIEWER COMMENTS

Reviewer #1 (Remarks to the Author):

The authors addressed all my comments. The addition of the explanation for the use of Zr really adds a lot to the paper, and the new results and figures are well detailed. They also provided interesting ideas moving forward to develop these membranes for higher performance. I agree with the authors that complete, or near complete, coverage could be possible. Challenging, but possible. Finally, all the materials and methods are now complete, which is good.

I do not have any further comments. I believe this is a very interesting work and I am looking forward to how these vertically aligned membranes will evolve.

Reviewer #2 (Remarks to the Author):

Although the authors provided responses to my previous comments, most of my major concerns, such as the lack of thorough material characterization data to sustain their claims, have not been well addressed. Moreover, there are two additional flaws in their revised manuscript:

1. Line 207 to 217, the authors claimed that "the free spacing of Zr-GO nanochannels can be calculated as 3.5 Å." However, such a claim is not consistent with their XRD data. In particular, XRD patterns (Supplementary Fig. 4b) indicate the d-spacing of 8.8 Å. After reducing the thickness of ~3.0 Å for one GO nanosheet, the free-spacing between the nanosheets (i.e., the channel height for mass transport) should be 5.8 Å, which is larger than the size of 2-propanol (~4.9 Å). In this case, 2-propanol vapor should transport through the GO film, which is contradictory to the authors' results (Figure 3b).

2. Even we assume their calculation was current and the free spacing is indeed 3.5 Å, such data still could explain their newly added liquid filtration data. The author claimed that "VAGME films show the expected high selectivity for Na(I) over the larger Mg(II) that is characteristic of GO nanochannel transport." The author further explained that: "The permeation rate of NaCl is two orders of magnitude higher than that of MgCl₂, which is consistent with the hydrated diameter difference between Na(I) and Mg(II)." These analyses indicate that their GO films exhibited much better rejection of NaCl than MgCl₂. Nevertheless, it should be emphasized that Na⁺ and Mg²⁺ ions have a hydration diameter of ~7.2 Å and 8.5 Å, respectively (Abraham, J. et al. Nat. Nanotechnol. 2017), such that both ions should be effectively rejected by their GO channels with the free spacing of 3.5 Å. Consequently, I would think there were some major flaws in their filtration test design or data interpretation. Additionally, the authors indicate that Na⁺ has a hydration diameter of 4.5 Å in Supplementary Table 4, which is another major mistake in the manuscript.

Again, these major concerns remarkably lower the reliability of their analysis and the overall quality of the study.

Responses to Reviewer Comments

We are pleased to submit a revised version and set of responses to this second set of reviewer comments (below). We have included 5 new figure panels and significant new text passages and believe we have been able to address all of the reviewer requests for additional clarification on both the material characterization and the interpretation of interlayer spacings for nanochannel transport.

Reviewer #1

Comment: The authors addressed all my comments. The addition of the explanation for the use of Zr really adds a lot to the paper, and the new results and figures are well detailed. They also provided interesting ideas moving forward to develop these membranes for higher performance. I agree with the authors that complete, or near complete, coverage could be possible. Challenging, but possible. Finally, all the materials and methods are now complete, which is good.

I do not have any further comments. I believe this is a very interesting work and I am looking forward to how these vertically aligned membranes will evolve.

Response: We appreciate the reviewer's positive response to the manuscript.

Reviewer #2

Comment: Although the authors provided responses to my previous comments, most of my major concerns, such as the lack of thorough material characterization data to sustain their claims, have not been well addressed.

Response: Based on reviewer's comment, here we provide a more thorough characterization and related discussion on the assembly and alignment of nanosheets in the original GO and Zr-GO

films. This includes five new Figure panels in the Supporting Information and significant new text passages in both the main paper and the SI. In addition, the structures of both the planar and wrinkled GO films have been well characterized in previous studies, and our new text now provides a set of literature references for readers interested in learning more about the structure of GO nanosheet films in either the planar or wrinkled states. Between this manuscript and the related literature, these films have been characterized extensively using the full set of standard relevant structural diagnostic tools in the graphene film field.

Content added to the main text:

“The high magnification view (Fig. 2b and Supplementary Fig. 4-5) of an individual Zr-GO strip shows a tightly packed, multilayered structure, which appears identical to original planar Zr-GO films. More information on the assembly and alignment of nanosheets in both the original GO and Zr-GO films can be found in the Supplementary Note 2 and references therein.”

Content added to the SI:

“As shown in Supplementary Fig. 4a and 1a, the GO nanosheets are $\sim 1 \mu\text{m}$ in lateral size and 1 nm in thickness, with a relatively broad distribution in lateral dimension, as is typical of exfoliated materials. During drying induced assembly, GO nanosheets align and stack by volume exclusion effects into an irregular tiling pattern to form a paper-like macroscopic structure (Supplementary Fig. 4b-c) where nanosheets are held together by hydrogen bonding, van der Waals forces, and π - π interactions in unfunctionalized regions, providing strong adhesion within the layers of the lamellar paper-like structure⁷⁻⁹. Such GO films therefore possess significant mechanical strength, which enables the fabrication of various GO multilayered structures, including wrinkled and crumpled films, and hierarchical deformed topographies¹⁰⁻¹². The cross section of a GO planar film in Supplementary Fig. 4c exhibits well-ordered layer packets, in which each packet consists of ~ 7 stacked GO nanosheets⁷. The ordered packets of nanosheets in GO films remain during deformation and realignment (Supplementary Fig. 4d).

Supplementary Figure 4. Assembly and alignment of GO nanosheets. **a**, Top view of individual GO nanosheets imaged by drop casting GO dilute suspension on a silicon substrate and SEM. **b**, Top view of a planar GO film assembled from GO nanosheets, which are smooth except for fine wrinkle structures that form spontaneously upon drying. **c**, Cross-sectional view of 1 μm thick GO planar film. **d**, Top view of a wrinkled GO film by unidirectional compressions of a planar GO film, showing the smooth continuous nature of the film surface. Scale bar, 1 μm .”

{Note to reviewer: It is not normally possible to visualize individual nanosheets in a multilayer film of this type and thickness. One can see stacking orientations as patterns in cross section (edge views), but the top surfaces are smooth and nearly featureless. These films are also opaque to TEM. Some sense for alignment and stacking of individual nanosheets can be gained by looking at sparse (i.e. thin and not fully dense) films specially prepared by deposition on Si metal substrates. We now provide one of these in Fig. S4a, and it gives a sense for the alignment and random tiling of individual nanosheets in typical multilayer films.}

More new content in the SI:

“Similarly, casting Zr-GO nanosheet suspensions onto substrates produces intercalated laminated films as shown by time-resolved XRD in Supplementary Fig. 5a. During the drying process, the Zr-GO nanosheets slowly stack together and form a lamellar structure, exhibiting an interlayer spacing of 8.8 angstroms. After realignment into VAGME, the GO strips still exhibit multi-layered structures as shown in Supplementary Fig. 5b. From a fractured VAGME (Supplementary Fig. 5c) we can see the tilted Zr-GO strips attached on the side of the epoxy, and the fine, irregular kinks in the film (in the X direction in Fig. 5b) appear now as near-vertical ridges in the fracture surface (Fig. 5c), which also reflect the near-vertical alignment of the nanochannels within the portion of the film strip attached to the epoxy.

Supplementary Figure 5. Assembly of Zr-GO nanosheets in VAGME. **a**, Time-resolved XRD tracking the appearance of lamellar structures in Zr-GO films during drying. **b**, Top view of Zr-GO strips on the VAGME surface, exhibiting multi-layered structure. **c**, Side view of a fractured VAGME, exposing the exfoliated Zr-GO short films attached on the side of epoxy matrix. Scale bar, 2 μm .”

{Note: panel c is new}

Comment: Moreover, there are two additional flaws in their revised manuscript:

1. Line 207 to 217, the authors claimed that “the free spacing of Zr-GO nanochannels can be calculated as 3.5 Å.” However, such a claim is not consistent with their XRD data. In particular, XRD patterns (Supplementary Fig. 4b) indicate the d-spacing of 8.8 Å. After reducing the thickness of ~ 3.0 Å for one GO nanosheet, the free-spacing between the nanosheets (i.e., the channel height for mass transport) should be 5.8 Å, which is larger than the size of 2-propanol

(~4.9 Å). In this case, 2-propanol vapor should transport through the GO film, which is contradictory to the authors' results (Figure 3b).

2. Even we assume their calculation was correct and the free spacing is indeed 3.5 Å, such data still could explain their newly added liquid filtration data. The author claimed that “VAGME films show the expected high selectivity for Na(I) over the larger Mg(II) that is characteristic of GO nanochannel transport.” The author further explained that: “The permeation rate of NaCl is two orders of magnitude higher than that of MgCl₂, which is consistent with the hydrated diameter difference between Na(I) and Mg(II).” These analyses indicate that their GO films exhibited much better rejection of NaCl than MgCl₂. Nevertheless, it should be emphasized that Na⁺ and Mg²⁺ ions have a hydration diameter of ~7.2 Å and 8.5 Å, respectively (Abraham, J. et al. Nat. Nanotechnol. 2017), such that both ions should be effectively rejected by their GO channels with the free spacing of 3.5 Å. Consequently, I would think there were some major flaws in their filtration test design or data interpretation.

Response: These are insightful comments and are related to molecular theories of nanochannel transport in the literature. Below we provide a detailed response and describe our revisions.

We would first like to say that the present manuscript is not attempting to offer a new theory of selective nanochannel transport. Our goal rather is to demonstrate that our membranes provide the same, known selectivity characteristics already reported in the literature to confirm that the nanochannels are preserved after re-alignment and fixation in a thin membrane and are not altered or destroyed by the processing, or made irrelevant by the presence of defects introduced by our processing. To that end, we can report that our membranes show the same selectivity for Na⁺ over Mg²⁺ (namely Na⁺ flux >> Mg²⁺ flux) as measured by Abraham et al. [Nature Nano, 2017] and H₂O >> isopropanol as reported by Liu et al., [Carbon 77 933–938 (2014)]. In the vapor phase they show selective transport of H₂O >> hexane as reported in Nair et al., [Science, 2012], and Spitz-Steinberg et al., [ACS Nano, 2017)]. These are experimental facts that confirm our membranes have transport properties characteristic of GO nanochannels (and not characteristic of larger pores associated with possible membrane flaws). This confirmation does not depend on any particular theory of molecular behavior inside nanochannels.

That said, we are happy to discuss the molecular origins of these effects, which readers may find interesting. The reviewer is correct that the d-spacing of Zr-GO films is measured here to be 0.88 nm, and if one subtracts the thickness of a graphene layer (usually taken as 0.34 nm, the interlayer spacing of typical graphene-based carbon materials) one gets 0.54 nm. Indeed this is slightly larger than the kinetic diameter of isopropanol (0.49 nm), but 0.49 nm is for the bare molecule in the gas phase (a “kinetic diameter”), while isopropanol is being transported through GO interlayers that are still hydrated. Our GO gallery spaces in these experiments are still nearly saturated with H₂O. IPA has an H-bond donor and acceptor site and would be hydrated during transport in those spaces, making it larger. Overall, it is fully expected that permeation of such a molecule would be slow through GO films, and also reported experimentally by Liu et al., 2014 (see reference above).

Secondly, it is already known that the simple permeation criterion based on comparing hydrated molecular diameter to the “free spacing” estimated as (Free Space = Interlayer-Space - 0.34 nm) is not fully predictive. It gives an indication worthy of discussion, but not a quantitative prediction. Here is a supporting passage from the SI of Abraham, J. et al. [Nat. Nanotechnol. 2017]:

“The hydrated diameters considered for all the ions in Fig. 2 of the main text are obtained from Ref. (2). There are large variations in exact values of hydrated diameters reported in literature³, due to disparities in the definition and differences in modelling parameters. For example, the reported hydrated diameter of K⁺ varies from 4 to 6.6 Å and for Mg²⁺ it varies from 6 to 9.4 Å. The chosen values in the main Fig. 2 are 6.6, 7.1, 7.6, 8.2 and 8.5 Å for K⁺, Na⁺, Li⁺, Ca²⁺ and Mg²⁺ respectively. However, irrespective of the chosen hydrated diameter, the absence of a pure size exclusion mechanism in the ion permeation through PCGO membrane is clear. **For example, the smallest reported hydrated diameter for Na⁺ ion is 5.4 Å, so it is not expected to permeate through PCGO membranes with an interlayer spacing smaller than 8.8 Å if the permeation cut-off is dictated by the size exclusion. The observed permeation of Na⁺ through this membrane confirms that ion permeation through PCGO membranes is not exclusively limited by their hydrated diameter.”**

One reason for the lack of predictive ability is that a portion of the hydration shell may be removed when the ion enters the channels, which is a phenomenon that also occurs in biological ion channels. The selectivity for Na^+ over Mg^{2+} may be due to the smaller hydrated diameter for Na^+ and/or its larger crystal radius and smaller charge, which weakens the hydration shell, making it easier to partially detach when entering or transporting through nanochannels (Tansel, B., et al [Sep. Purif. Technol., 2006]; Tansel, B. [Sep. Purif. Technol., 2012]).

Finally, there are competing theories about the estimation of “free space” in GO nanochannels. Subtracting only the graphene layer thickness (0.34 nm) makes sense if the channels are a connected network of unfunctionalized regions lying between oxidized domains. Recent literature suggests that, at least in some materials, the oxidized domains are the continuous phase and the pristine regions are isolated. In such a case, transport must occur through the oxidized regions, whose spacing must be estimated by also subtracting the size of the out-of-plane functional groups. This was the conceptual model used in the discussion in our manuscript. This model may be more appropriate for materials with high oxygen contents and is supported by recent studies using aberration corrected TEM (Erickson, K. et al [Adv. Mater., 2010]; Pacilé, D. et al [Carbon, 2011]). This model gives somewhat different estimates of free spacing, but these can also rationalize the experimental trends (with the caveats given above about relying too much on quantitative comparisons with hydrated diameters). The real claims in our paper do not depend on these molecular theories, or depend on one or the other being correct. We have revised the paper to give more details and information on both models for nanochannel size estimation (see below).

Comment: Additionally, the authors indicate that Na^+ has a hydration diameter of 4.5 Å in Supplementary Table 4, which is another major mistake in the manuscript.

Response: We would like to point out that there are large variations in the size of hydrated ions reported in the literature. For Na^+ in particular, we find values from 0.45 - 0.72 nm. The determination of hydration number can vary depending on which method was used, resulting in a spread of estimated values. Our hydrated diameters of cationic ions (Na^+ , K^+) were adopted from

Kielland, J., Individual activity coefficients of ions in aqueous solutions, *J. Am. Chem. Soc.* 59:1675-1678, 1937. It provides a thorough estimation of hydrated size on 130 inorganic and organic ions, thus is actively cited by 1844 times up to now. For example, it was cited in N. A. Kotov et al., *Science* 350, 1242477 (2015) for Na⁺: “The diameters of common ions with their hydration shells encountered in NP dispersions are, for instance, 0.3 to 0.64, 0.45 to 0.47, 0.46 to 0.5, 0.40 to 0.6, 0.34 to 0.8, and 0.7 to 1.1 nm for Cl⁻, Na⁺, Cd²⁺, Ca²⁺, Mg²⁺, and CO₃²⁻, respectively (30–32); these values are comparable to the diameters of many small NPs.” Again the SI of Abraham, J. et al. [*Nat. Nanotechnol.* 2017] also states that there are large variations in exact values of hydrated diameters reported in literature³, due to disparities in the definition and differences in modelling parameters.

Within this range, the particular value we quote in the article has no special significance in the interpretation of the data. We have revised the manuscript to provide information on the range of values reported, to give readers a more accurate picture of the field.

Below is a listing of the changes we made to the text:

Content added to the main text:

“In general, water molecules enter nanochannels driven by capillary forces, and are reported to flow preferentially through non-oxidized regions^{3,12,28}. The size of these channels can be estimated by subtracting the width of a pristine (non-oxidized) graphene layer (0.34 nm) from the XRD interlayer spacing (here 0.88 nm) to give 0.54 nm. Some transport processes may occur through the oxidized regions, where channel sizes are smaller due to out-of-plane oxygen-containing functional groups^{15,29,30} and can be estimated here as 0.34 nm (see detailed discussion in Supplementary Note 3). Both types of channels are expected to reject 2-propanol molecules, which have a (bare) kinetic diameter of 0.49 nm³¹ and are expected to be larger within the water-saturated channels in these membranes due to hydration at the site of the H-bond active hydroxyl group.”

Content added to the SI:

“In general, GO possesses oxidized and non-oxidized regions, where oxygen-containing groups function as spacers between nanosheets. Water molecules enter nanochannels driven by capillary forces, and are reported to experience frictionless flow through non-oxidized regions or slower flow through oxidized regions^{12,13}. The size of the non-oxidized channels can be estimated by subtracting the width of a pristine (non-oxidized) graphene layer (0.34 nm) from the XRD interlayer spacing (here 0.88 nm) to give 0.54 nm. In some materials with high oxygen content, the oxidized regions form the continuous network, leaving non-oxidized regions as isolated islands^{14,15}. In these cases, molecular transport in GO membranes may be forced to occur primarily through the oxidized regions. The sizes of those channels can be estimated here as 0.88 nm (measured d-spacing of Zr-GO films) – 0.34 nm (graphene layer thickness) – 0.2 nm (functional group size) = 0.34 nm, for alternative estimate of the relevant channel size for molecular exclusion phenomena in this study¹⁶⁻¹⁸.”

Additional content added to the main text:

“Beyond vapor permeation, liquid filtration tests on NaCl and MgCl₂ aqueous solutions were also conducted and discussed. More detailed discussion is included in the Supplementary Note 3.”

Content added to the SI:

“All liquid filtration tests were done using the device shown in Supplementary Fig. 9. Permeation data for NaCl and MgCl₂ aqueous solutions through VAGME are shown in Supplementary Fig. 10 and Supplementary Table 4. The permeation rate of NaCl is two orders of magnitude higher than that of MgCl₂, consistent with data from Abraham, J. et al²³. The selectivity for Na⁺ is likely due to its smaller hydrated diameter and/or its larger crystal radius and smaller charge, which weakens the hydration shell, making it easier to partially detach when entering or transporting through nanochannels²⁴⁻²⁶.”

Supplementary Table 4. Permeation rates of ions through VAGME and the hydrated diameter of metal ions.

Ions	Permeation rate mol m ⁻² hr ⁻¹	Hydrated diameter ^{24,26} nm
Na(I)	1.1±0.1	0.45-0.72
Mg(II)	(1.9±0.2)×10 ⁻²	0.60-0.94

”

REVIEWERS' COMMENTS

Reviewer #2 (Remarks to the Author):

Revision is adequate

Responses to Reviewer Comments

Reviewer #2

Comment: Revision is adequate.

Response: We appreciate the reviewer's positive response to the manuscript.